# An Overview of the Structure–Activity Relationship in Novel Antimicrobial Thiazoles Clubbed with Various Heterocycles (2017–2023)

**DOI:** 10.3390/pharmaceutics16010089

**Published:** 2024-01-09

**Authors:** Daniel Ungureanu, Brîndușa Tiperciuc, Cristina Nastasă, Ioana Ionuț, Gabriel Marc, Ilioara Oniga, Ovidiu Oniga

**Affiliations:** 1Department of Pharmaceutical Chemistry, Faculty of Pharmacy, “Iuliu Hațieganu” University of Medicine and Pharmacy, 41 Victor Babeș Street, 400012 Cluj-Napoca, Romania; daniel.ungureanu@elearn.umfcluj.ro (D.U.); cmoldovan@umfcluj.ro (C.N.); ionut.ioana@umfcluj.ro (I.I.); marc.gabriel@umfcluj.ro (G.M.); ooniga@umfcluj.ro (O.O.); 2“Prof. Dr. Ion Chiricuță” Oncology Institute, 34-36 Republicii Street, 400015 Cluj-Napoca, Romania; 3Department of Pharmacognosy, Faculty of Pharmacy, “Iuliu Hațieganu” University of Medicine and Pharmacy, 12 Ion Creangă Street, 400010 Cluj-Napoca, Romania; ioniga@umfcluj.ro

**Keywords:** thiazole, structure–activity relationship, antimicrobial, heterocycles, hybrid compounds

## Abstract

Antimicrobial resistance is an increasing problem for global public health. One of the strategies to combat this issue is the synthesis of novel antimicrobials through rational drug design based on extensive structure–activity relationship studies. The thiazole nucleus is a prominent feature in the structure of many authorized antimicrobials, being clubbed with different heterocycles. The purpose of this review is to study the structure–activity relationship in antimicrobial thiazoles clubbed with various heterocycles, as reported in the literature between 2017 and 2023, in order to offer an overview of the last years in terms of antimicrobial research and provide a helpful instrument for future research in the field.

## 1. Introduction

Antimicrobial resistance (AMR) represents a major threat to global public health and is an increasing challenge to overcome. According to the latest statistics [1], AMR alone has caused 1.27 million deaths globally, with the indiscriminate use of antimicrobials, not only for humans but also for livestock, representing the main cause of the problem. Pathogens with the most increased risk for developing resistance to carbapenems and to new molecular entities of newer generation cephalosporins are *Pseudomonas aeruginosa*, *Acinetobacter baumanii*, and Enterobacteriales [1].

Strategies in medicinal chemistry for fighting AMR include the total synthesis of well-established antibiotic classes based on extensive structure–activity relationships of preexistent antibiotic classes (for example, eravacycline); the optimization of the physicochemical property space (finafloxacin); the knowledge of the target in order to design new scaffolds (avibactam and vaborbactam); the synthesis of hybrid compounds with synergistic activity (cadazolid); the re-exploration of the targets of older antibiotics (griselimycins targeting DnaN, the sliding clamp of DNA polymerase); and the targeting of virulence factors (elastase B, lectins A and B of *P. aeruginosa*, quorum-sensing, or transcriptional regulator PqsR) [2].

Thiazole is a five-membered heterocycle, part of the azoles group containing one nitrogen heteroatom and one sulfur heteroatom. It is found in a series of authorized antimicrobial drugs, such as aztreonam, and various cephalosporins (including the newest cefiderocol), abafungin, isavuconazole, ravuconazole, ethaboxam, and myxothiazol, thus underlining the importance of this nucleus for antimicrobial activity [3,4,5] (Figure 1). The effect of the thiazole nucleus on the biological activity of a molecule can be influenced by clubbing it directly or through a linker with other moieties with pharmacological potential, thereby creating hybrid compounds [6].

Knowledge of qualitative and quantitative structure–activity relationship (SAR) studies is of great importance in a rational drug design. Therefore, this review aims to study the structure–activity relationships of novel antimicrobial thiazole clubbed with various heterocycles in hybrid compounds, as reported in the last seven years, as a helpful tool for further research and the development of antimicrobials containing a thiazole heterocycle. The publications were selected by interrogating Scopus, SpringerLink, PubMed, and EMBASE databases, using “Thiazole AND Hybrid AND Antimicrobial” as the search syntax, filtering the results by the “Classical Article” filter as the article type, and setting the publication date range between 2017 and 2023. A total of 427 results from all four databases were obtained. However, it is important to mention that many of the results were indexed in two or more databases. A bibliometric analysis performed of the selected publications revealed that thiazole in antimicrobial hybrid compounds is a topic of increasing interest in the last four years (Figure 2). Therefore, 50 publications that met the eligibility criteria for this review (synthesis of hybrid compounds bearing the thiazole scaffold clubbed with other heterocycles) were discussed.

## 2. Structure–Activity Relationships in Antimicrobial Thiazole-Based Hybrid Compounds

The selected papers were organized based on the nature of heterocycles that were clubbed with thiazole. Additionally, they were ordered by their size, starting with the smallest heterocycles and finishing with polyheterocyclic systems (Figure 3). This offered an overview of how antimicrobial activity may also be affected by this aspect. It is worth mentioning that no research on antimicrobial thiazoles clubbed with three-membered heterocycles was available in the literature in the specified timeframe. Thus, classification started with the four-membered group of heterocycles.

### 2.1. Thiazole Clubbed with Four-Membered Heterocycles

Four-membered heterocycles are a class of highly reactive and low-stability chemical entities, classified into unsaturated heterocycles—azetes, oxetenes, thietes—and their saturated derivatives—azetidines, oxetanes, and thietanes. The best known and important derivative of this class is 2-azetidinone. The chemical instability of four-membered heterocycles makes it difficult to use them extensively in medicinal chemistry, hence the reduced number of hybrid compounds available in the literature. Herein, we present SAR studies of some antimicrobial thiazoles clubbed with 2-azetidinone hybrid compounds.

#### Thiazolyl–Azetidin-2-One Hybrid Compounds

2-Azetidinone is the centerpiece of β-lactam antibiotics and is also found in some β-lactamase inhibitors, like clavulanic acid, sulbactam, and tazobactam, thus making it an essential scaffold in the development of new potential antibacterial molecules [12].

Desai et al. reported the design and synthesis of a series of 4-(quinolin-3-yl)-1-(thiazol-2-yl)-amino-azetidin-2-ones (Figure 4) with two points of variation: the sixth position of the quinoline ring (R^1^) and the phenyl from the fourth position of the thiazole ring (R^2^) [13]. Depending on each case, the quinoline ring can be unsubstituted (**1**–**3**) or substituted with a methyl group (**4**–**7**) in the sixth position. Similarly, the thiazole ring can be either unsubstituted (**4**) or substituted with electron-donating (**1**, **5**, and **7**) or electron-withdrawing groups (**2**, **3**, and **6**) in the fourth position [13].

The compounds were tested for their antibacterial activity against Gram-positive and Gram-negative bacteria and for their antifungal activity against *Candida* sp. and *Aspergillus* sp. strains. The results were quantified as minimal inhibitory concentrations (MICs) [13]. Overall, the compounds were more effective against the Gram-negative strains compared to the Gram-positive ones. Compounds **1**–**4** and **6** showed the strongest antibacterial activity against *E. coli* MTCC 443 (MICs = 100 µg/mL), while compound **5** was the most efficient against *P. aeruginosa* MTCC 1688 (MIC = 100 µg/mL), but, in both cases, the activity was inferior to chloramphenicol (MIC = 50 µg/mL) [13].

In terms of antifungal activity, compounds **5** and **7** showed inferior activity (MIC = 200 µg/mL) against *Candida albicans* MTCC 227, *Aspergillus niger* MTCC 282, and *A*. *clavatus* MTCC 1323, compared to nystatin (MIC = 100 µg/mL) [13].

SAR studies of this series showed that R^2^ substitution was associated with increased antibacterial activity (Figure 4). Electron-donating groups for both R^1^ and R^2^ (**5** and **7**) were associated with antifungal activity [13].

The compounds are potential inhibitors of β-ketoacyl-acyl-carrier protein synthase III, which is an important enzyme in fatty acid biosynthesis and the growth of bacteria, according to molecular docking studies. The proposed target has not been verified in a biological assay. No potential mechanism for antifungal activity was reported [13].

### 2.2. Thiazole Clubbed with Five-Membered Heterocycles

Five-membered heterocycles represent an extensive group of valuable structures for designing pharmacologically active compounds, including antimicrobials. Compared to four-membered heterocycles, they have significantly greater chemical stability and versatility, which is transposed in their ubiquitarian usage in pharmaceutical research and development.

Based on the literature search of the last seven years, we will further discuss structure–activity relationships in antimicrobial hybrid compounds containing thiazole clubbed with the following five-membered heterocycles and their derivatives: pyrazoline, pyrazolinone, pyrazole, imidazole, thiazolidinone, thiazolidindione, 1,3,4-thiadiazole, 1,2,3-triazole, 1,3,4-oxadiazole, and thioxo-1,3,4-oxadiazole.

#### 2.2.1. Thiazolyl–2-Pyrazoline Hybrid Compounds

2-Pyrazoline is an important scaffold in anti-infective drugs, possessing antibacterial, antifungal, antiviral, antiparasitic, and antituberculosis potential [14]. Herein, we present the structure–activity relationship for some series of thiazole linked with 2-pyrazoline derivatives with promising antimicrobial potential to establish how clubbing these two heterocycles influences the biological activity of the entire molecule.

Based on the structures found, it is possible to conclude that there is one general scaffold consisting of both heterocycles linked directly but in different positions. Therefore, three types of linking patterns have been identified: the fifth position of the thiazole ring to the fifth position of the 2-pyrazoline ring [15], the second position of the thiazole ring to the first position of the 2-pyrazoline ring [16,17,18,19,20,21,22,23], and the fifth position of the thiazole ring to the third position of the 2-pyrazoline ring [24] (Figure 5).

Cuartas et al. designed four series (**8a**–**14a**, **8b**–**14b**, **8c**–**14c**, **8d**–**14d**) of 2-(*N-*mustard)-5-(2-pyrazolin-5-yl)-thiazoles (Figure 6) with two points of variation: the substituted phenyl from the third position of the 2-pyrazoline (R) and the first position of the 2-pyrazoline ring (R^1^) [15]. The compounds bore a nitrogen mustard moiety, known for its DNA alkylating properties. Depending on each case, R^1^ could be a carbonylic group (**a** and **b**) or a phenyl ring (**c** and **d**), while R could be hydrogen (**14**), a halogen atom (**8**–**10**), or etheric groups (**11**–**13**).

The compounds were tested for their antifungal activity against *Candida* sp. and *Cryptococcus* sp. strains and for their antibacterial activity against Gram-positive and Gram-negative strains. The results were quantified as IC_50_, representing the minimum concentration that inhibits 50% of growth and MIC [15]. Compounds **8c**–**14c** (R^1^ = C_6_H_5_) and **12d**–**14d** (R^1^ = 3,5-di-Cl-C_6_H_3_) showed inferior activity (IC_50_ = 15.6–125 µg/mL) against *Cryptococcus neoformans* ATCC 32264 compared to amphotericin B (IC_50_ = 0.50 µg/mL) [15].

In terms of antibacterial activity, the compounds showed significant effects against vancomycin-intermediate *S. aureus* (VISA) (MICs = 3.25–500 µg/mL), methicillin-susceptible *S. aureus* (MSSA ATCC 25923) (MICs = 62.5–500 µg/mL), methicillin-resistant *S. aureus* (MRSA ATCC 43300) (MICs = 125–500 µg/mL), and *N. gonorrhoeae* ATCC 31426 (MICs = 125–500 µg/mL), compared to penicillin, ceftriaxone, and ciprofloxacin [15].

SAR studies revealed that the nature of the substituents on 2-pyrazoline (**a**–**d**) and phenyl (**8**–**14**) were the most important for antibacterial and antifungal activities (Figure 6) [15]. The phenyl substituent from the first position of the 2-pyrazoline ring (**c**) conferred anticryptococcal activity regardless of the other substituents (**8c**–**14c**), while the 3,5-dichlorophenyl substituent (**d**) was conditioned by electron-donating groups, particularly methoxy (**12d**) or trimethoxy (**13d**), or no substituent at all (**14d**) on the phenyl ring from the third position of the 2-pyrazoline ring. It is worth mentioning that the 3,5-dichlorophenyl substituent is found in the structure of some important antifungal azoles like miconazole, ketoconazole, or itraconazole. However, the situation is the opposite for antibacterial activity. The 3,5-dichlorophenyl substituent abolished the antibacterial effect, while, for phenyl-substituted derivatives, it was very low. The best activity was observed when small substituents in R_1_ were present, like acetyl (**a**) and formyl (**b**). Another important aspect is that the presence of chloro (**8a** and **8b**), bromo (**9a** and **9b**), and fluoro (**9c**) substituents or no substituents (**14b**) increased activity against VISA (Figure 6) [15]. No potential target was reported by the authors.

Rashdan and Abdelmonsef designed 2-(4-(1-thiazol-2-yl)-2-pyrazolin-3-yl)-1,2,3-triazol-1-yl)-1,3,4-thiadiazole with a single point of variation: the fifth position of the 2-pyrazoline ring (R) (Figure 7) [16].

These compounds were assayed for their antibacterial activity against Gram-positive and Gram-negative bacteria and for their antifungal activity against *C. albicans* [16]. The thiophene-substituted compound **15** showed promising results against *E. coli*, *P. aeruginosa*, and *S. aureus* (MICs = 5–10 µg/mL), compared to ciprofloxacin (MICs = 1.25–7 µg/mL) [16].

Concerning antifungal activity, the same compound showed identical activity to nystatin against *C. albicans* (MIC = 5 µg/mL) [16].

Based on the minimal structural differences between the derivatives obtained, it seems that switching from a bulkier (3-metoxy-4-hydroxy-phenyl in **16**) to a smaller substituent (2-thienyl in **15**) enhanced the activity (Figure 7) [16].

Additionally, compound **15** was a potent inhibitor of SARS-CoV-2 transmembrane serine protease 2 (TMPRSS2), which plays an important role in disease propagation, based on molecular docking studies. The proposed target has not been verified in a biological assay [16].

Budak et al. reported the synthesis of a series of 2-(4-(1-(thiazol-2-yl)-2-pyrazolin-3-yl)-phenyl)-methanoisoindol-1,3-dione derivatives. The compounds obtained varied by the substituent in the fifth position of the 2-pyrazoline ring (R), which could be an aryl (**17**–**23**) or hetaryl ring (**24**–**25**) (Figure 8) [17].

The compounds were tested for their antimicrobial activity against Gram-positive and Gram-negative bacteria and *C. albicans* ATCC 1213 [17]. The results were quantified by an inhibition zone (IZ) through a disk diffusion method. All of the compounds (**17**–**25**) showed inferior activity against *S. aureus* ATCC 29213 (IZs = 10–19 mm), compared to cefoperazone–sulbactam (IZs = 19–26 mm). Additionally, compound **25** was active against *Proteus vulgaris* KUEN 1329 (IZ = 10 mm) but inferior to the standard drug [17].

SAR studies of this series showed that aryl substitution induced better activity compared to hetaryl substitution (Figure 8). Thus, in the aryl-substituted derivatives, para-substitution (**17**–**20**) favorized overall better activity compared to meta-substitution (**21**–**23**). The position of the R^1^ substituent was found to be more important than their nature, as substitution in para with both electron-withdrawing and electron-donating groups yielded similar activities [17]. In the case of compounds **24** and **25**, bearing 2-thienyl and 2-furanyl substituents, a broader activity spectrum was observed for **24**, which could be attributed to the higher aromaticity of 2-thienyl compared to 2-furanyl substituent, making it suitable for the bioisosteric substitution of phenyl rings [17]. No potential target was reported by the authors.

Mansour et al. designed three series of 2-(3-aryl-5-hetaryl-2-pyrazolin-1-yl)-thiazole derivatives (Figure 9) with three points of variation: one is the linking position to the naphthyl ring (**a** = 1-naphthyl or **b** = 2-naphthyl) from the third position of the 2-pyrazoline ring and the other two are in the fourth (R^1^) and fifth positions (R^2^) of the thiazole ring [18]. Depending on each case, R^1^ can be a para-halogen-substituted phenyl ring (**26** and **27**) or a methyl group (**28**–**33**), while R^2^ can be either hydrogen (**26** and **27**) or various acyl substituents (**28**–**30**) and arenediazo groups (**31**–**33**) (Figure 8) [18].

The compounds were tested for antimicrobial activity against Gram-positive and Gram-negative bacteria and various fungal strains [18]. Most of the compounds displayed antibacterial activity against *S. aureus* (IZs = 0.5–2.6 mm) and antifungal activity against *A. flavus* (IZs = 0.5–2.3 mm) but were inferior compared to amoxicillin (IZs = 2.2–3.5 mm) and griseofulvin (IZs = 2.1–3.3 mm). Only the compounds **27b** and **33a** were active against all the tested microbial strains (*S. aureus*, *Bacillus subtilis*, *K. pneumoniae*, *P. aeruginosa*, *A. fumigatus*, *A. flavus*, *Syncephalastrum racemosum*, *Penicillium expansum*, and *C. albicans*) [18].

SAR studies suggest the activity depends on the nature of R^1^ and R^2^ and the linkage of the 2-pyrazoline ring to the naphthyl group [18]. Thus, in the 4-(*p*-halophenyl)-thiazolyl series (**26a,b** and **27a,b**) (Figure 9), the bromo substituent (**27a** and **b**) inactivated the compounds, while the chloro substituent (**26a** and **b**) was responsible for the antibacterial activity, apart from compound **26b**, which displayed activity against all the tested bacterial and fungal strains. Therefore, chloro and 2-naphthyl substituents (**26b**) were the best combinations for highly active antimicrobials [18].

In the 5-acylthiazolyl series (**28a,b**, **29a,b**, and **30a,b**) (Figure 9), the compounds substituted with acetate (**29a** and **29b**) or anilido (**30a** and **30b**) groups were inactive or had a very low activity—only against *A. flavus* (IZ = 0.5 mm) and *P. expansum* (IZ = 0.8 mm)—while acetyl substitution (**28a** and **28b**) yielded moderate antimicrobial activity (IZs = 0.5–1.2 mm) [18].

Finally, in the 5-arendiazothiazolyl series (**31a,b**, **32a,b**, and **33a,b**), the activity depended more on the linkage to the naphthyl group (**a** or **b**) [18]. Thus, the best combination was between *p*-chlorobenzenediazo and 1-naphthyl (**33a**), which was the most active. Combinations between *p*-toluenediazo and 1-naphthyl (**32a**) or benzenediazo and 2-naphthyl (**31b**) yielded inactive molecules (Figure 9) [18]. No potential target was reported by the authors.

Using a similar scaffold, Masoud et al. [19] reported the synthesis of two more series of 2-(3-aryl-5-hetaryl-2-pyrazolin-1-yl)-thiazole derivatives, with two points of variation: (R) linked to the fifth position of the 2-pyrazoline ring (**c** = 3,4-dimethoxyphenyl and **d** = 1,3-benzodioxole) and (R^1^) linked to the fifth position of the thiazole ring [19]. Depending on each case, R^1^ could be various acyl substituents (**34**–**36**) or arenediazo groups (**37**–**39**) (Figure 9).

The antimicrobial effect was assayed using the same strains and positive controls mentioned by Mansour et al. [19]. The most notable results for antibacterial activity were obtained for compounds **36d**, **38d**, and **39c** against *S. aureus* (IZs = 0.9–1.2 mm) and compounds **34c**, **35c**, **37c**, **38c**, and **39c** against *K. pneumoniae* (IZs = 1.2–2.0 mm) and *P. aeruginosa* (IZs = 1.0–1.8 mm) [19].

In the case of antifungal activity, the most important results were obtained for compounds **37c**, **38c**, and **39d** against *C. albicans* (IZs = 1.3–2.1 mm); compound **38c** against *A. fumigatus* (IZ = 1.2 mm); and compound **38d** against *A. flavus* (IZ = 2.3 mm) [19].

SAR studies of these compounds suggest that acyl group substitution in position 5 of the thiazole ring (**34c,d**, **35c,d**, and **36c,d**) only resulted in antibacterial active compounds (Figure 9) [19]. Favorable substituents for the antibacterial activity were acetate (**34c** and **d**), anilido (**35c** and **d**), and acetyl (**36c** and **d**), which was the opposite compared to the compounds by Mansour et al. [19].

Arendiazo substitution of the thiazole ring (**37c,d**, **38c,d**, and **39c,d**) expanded the spectrum against both bacterial and fungal strains. In this case, the main difference in the potence was dictated by whether the substituent from position 5 of the 2-pyrazoline ring was 3,4-dimethoxyphenyl (**c**) or 1,3-benzodioxole (**d**), with the first being more active [19]. No potential target was reported by the authors.

When analyzing the SAR studies of the molecules synthesized by both authors (Figure 9), it seemed that the naphthyl group brought drawbacks to the compounds designed by Mansour et al., as its antimicrobial potential was reduced compared to the compounds designed by Masoud et al., who replaced it with a 3,4-dimethoxyphenyl moiety. Thus, a higher polarity should be considered when designing novel antimicrobial compounds [18,19].

Bhandare et al. reported the synthesis of some 2-(2-pyrazolin-1-yl)-thiazoles in which the fourth position of the thiazole ring was linked via a methylene bridge to various thiol- and thioether-containing azoles (1,3,4-oxadiazole and 1,2,4-triazole) (Figure 10) [20].

The compounds were tested for their antibacterial activity against Gram-positive and Gram-negative strains and for their antifungal activity against *Candida* sp. and *Aspergillus* sp. strains [20]. Overall, all of the compounds displayed significant antimicrobial potential. Compounds **42**, **46**, **48**, and **49** showed similar activity (MICs = 0.5–8 µg/mL) against *S. aureus* ATCC 11632, *S. faecalis* ATCC 14506, *K. pneumoniae* ATCC 10031, *E. coli* ATCC 10536, and *P. aeruginosa* ATCC 10145, compared to ciprofloxacin (MICs between ≤1 and >5 µg/mL) [20].

Regarding antifungal activity, the same compounds showed similar activity against *C. tropicalis* ATCC 1369 and *A. niger* ATCC 6275, compared to fluconazole (MICs ≤ 1 µg/mL) [20].

SAR studies pointed out that the heterocyclic tetrad induced promising overall antibacterial and antifungal activities (Figure 10) [20]. An additional aryl substituent (**42**–**46** and **48**–**51**) in the thioether series was essential for these activities. Switching to a hetaryl substituent, 3-pyridinyl (**47**), decreased the activity. The substituents of aryl were also an important factor in determining antibacterial and antifungal strengths. Thus, a small halogen substituent, like 4-fluoro (**46** and **49**), induced excellent activity, while larger substituents, such as 2-trifluoromethyl (**44**), 4-trifluoromethoxy (**45**), 4- or 5-chloro (**43** and **50**), and 4-bromo (**51**), significantly decreased the activity. Of note, unsubstituted rings (**42** and **48**) also displayed important activity (Figure 10) [20].

These compounds could potentially target DNA gyrase—important for the replication of genetic material in bacteria—and Cytochrome P450 14 α-sterol demethylase—important for the synthesis of ergosterol from lanosterol—for antifungal activity, based on molecular docking studies. The proposed target has not been verified in a biological assay [20].

Abdel-Wahab et al. designed a series of 2-(5-(3-(1,2,3-triazol-4-yl)-pyrazol-4-yl)-2-pyrazolin-1-yl)-thiazole (Figure 11) [21]. These compounds are para-substituted on the phenyl ring from the third position of the 2-pyrazoline ring (R^1^) and have an additional substituent in the fourth position of the thiazole ring (R^2^).

The compounds were tested for their antimicrobial activity against Gram-positive and Gram-negative bacteria and *C. albicans* NRRL Y-477 [21]. Compounds **52** and **53** showed inferior activity (MIC = 50 µg/mL) against *S. aureus* ATCC 29213 and *K. pneumoniae* ATCC 13883, compared to ciprofloxacin (MIC = 25 µg/mL) [21].

Only compound **54** was active against *C. albicans* (MIC = 200 µg/mL), but its activity was inferior compared to clotrimazole (MIC = 25 µg/mL) [21].

SAR studies of these compounds suggest that the identic substitution (**52** and **53**) of both 2-pyrazoline and 1,3-thiazole moieties could be responsible for antibacterial activity (Figure 11). Compound **54**, containing an additional 1-(*p-*tolyl)-5-methyl-1,2,3-triazol-4-yl moiety, may induce a closer resemblance to fluconazole than the rest of the compounds by containing two triazole moieties in its structure, hence demonstrating better antifungal activity against *C. albicans* [21]. No potential target was reported by the authors.

Salih et al. synthesized two 2-(2-pyrazolin-1-yl)-thiazole compounds substituted in the fifth position of the 2-pyrazoline ring with a phenoxybenzyl moiety, which contains an electron-withdrawing group on the benzyl rest (**55** and **56**) (Figure 12) [22].

The compounds were tested for their antimicrobial activity against bacterial and fungal strains, using ciprofloxacin (MICs = 0.5–4 µg/mL) and fluconazole (MIC = 16 µg/mL) as references [22]. Compound **56** (MICs = 8–16 µg/mL) showed better antibacterial activity than compound **55** (MICs = 16–32 µg/mL) against *S. aureus*, *E. coli*, *P. aeruginosa*, and *A. baumanii*.

Concerning antifungal activity, both compounds showed equal activity against *C. albicans* (MIC = 32 µg/mL) but inferior activity to fluconazole [22].

Based on the results, substitution with 3-nitro (**56**) was more favorable for the overall antimicrobial activity compared to 4-chloro (**55**) substitution (Figure 12) [22].

These compounds can potentially target tyrosyl-tRNA synthetase (important for bacterial protein synthesis) and type IIA topoisomerase for antibacterial activity and sterol 14-alpha demethylase for antifungal activity, according to molecular docking studies. The proposed targets have not been verified in a biological assay [22].

Dawood et al. designed some 2-(3-(thiophen-2-yl)-2-pyrazolin-1-yl)-thiazole derivatives variously substituted in the fifth position of the 2-pyrazoline ring (R^1^), as well as the fourth (R^2^) and the fifth positions (R^3^) of the thiazole ring with aryl or hetaryl substituents (Figure 13) [23].

The compounds were tested for their antimicrobial activity against bacterial strains and the *C. albicans* ATCC 10231 fungal strain [23]. Compounds **57**–**60** showed superior activity against *P. aeruginosa* ATCC 29853 (MICs = 15.625–31.25 µg/mL) compared to amoxicillin (MIC > 500 µg/mL). This activity was inferior (MICs = 62.5–125 µg/mL) against *E. coli* ATCC 25922 and *S. aureus* ATCC 25923 compared to the reference (MICs = 7.8–62.5 µg/mL) [23]. The compounds were also tested against sensitive *M. tuberculosis*, with compound **59** showing equal activity to isoniazid (MIC = 0.12 µg/mL) [23].

Concerning antifungal activity, the compounds showed superior activity (MICs = 3.9–62.5 µg/mL) against *C. albicans* compared to fluconazole (MIC = 250 µg/mL) [23].

SAR studies of these series showed that a heterogenous substitution of the 2-pyrazoline ring (R^1^ = 2,5-dimethoxyphenyl) was more favorable for the overall antimicrobial activity compared to the thiophene bisubstitution of this ring (Figure 13). Substitution with *p*-bromophenyl in the fourth position of the thiazole ring increased the antifungal and antituberculosis activities of compound **59**. The presence of an ethyl carboxylate substituent in the fifth position of the thiazole ring in compound **60** slightly decreased the overall antimicrobial activity compared to compounds **57**–**59** (Figure 13) [23].

The antituberculosis activity of these compounds can be attributed to the potential inhibition of DHFR, based on molecular docking studies. This target has been verified in a DHFR inhibitory efficiency assay, with compound **59** showing better inhibition (IC_50_ = 4.21 ± 0.13 µM) compared to trimethoprim (IC_50_ = 6.23 ± 0.05 µM). No potential target was reported for antibacterial and antifungal activities [23].

Bondock and Fouda designed some 2-(*N*-allyl)-5-(2-pyrazolin-3-yl)-thiazole derivatives with various substituents in the first position of the 2-pyrazoline ring (**60**–**65**) (Figure 14) [24].

The compounds were tested for antimicrobial activity against Gram-positive and Gram-negative bacteria and fungal strains [24]. All of the selected compounds showed similar activities (MICs = 0.03–7.81 µg/mL) against *S. pneumoniae* RCMB 010010 and *S. epidermidis* RCMB 010024, compared to ampicillin (MICs = 0.6–0.24 µg/mL). Compounds **60**, **62**, and **65** showed similar activity (MICs = 0.03–7.81 µg/mL) against *E. coli* RCMB 010052, *P. vulgaris* RCMB 010085, and *K. pneumoniae* RCMB 010093, compared to gentamycin (MICs = 0.03–1.95 µg/mL) [24].

In terms of antifungal activity, which was similar to amphotericin B (MICs = 0.12–7.81 µg/mL), compounds **61**–**63** and **65** showed results against *A. fumigatus* RCMB 02568 (MICs = 0.12–7.81 µg/mL), while only compounds **62** and **65** were active against *S. racemosum* RCMB 05922 (MICs = 0.24–7.81 µg/mL) [24].

According to SAR studies, the substitution of the 2-pyrazoline ring with a thiocarbamide group (**62**) favorized overall antibacterial and antifungal activities (Figure 14), while the phenylthiocarbamide group (**63**) only induced good activity against Gram-positive bacteria and *A. fumigatus* [24]. Substitution with a phenyl ring (**61**) induced a moderate antifungal activity (Figure 14). Further expansion of the molecule with a phenylthiazolyl moiety (**64** and **65**) yielded overall medium antibacterial and antifungal activities. The 4-fluorophenyl derivative (**65**) was more potent against *S. epidermidis* RCMB 010024, *P. vulgaris* RCMB 010085, *A. fumigatus* RCMB 02568, and *S. racemosum* RCMB 05922, compared to the compound with unsubstituted phenyl (**64**) [24]. No potential target was reported by the authors.

To conclude the results observed in the papers analyzed (Figure 15), clubbing thiazole with 2-pyrazoline to obtain novel antimicrobials should be considered when aiming for compounds active against Gram-positive bacterial strains, especially against *S. aureus*. Moderate results were observed against Gram-negative bacterial strains, except against *K. pneumoniae*, where the results were promising [18,19,20,21,24].

Antifungal activity was much lower compared to antibacterial activity. Thus, this scaffold may not be suitable for designing potent antifungals.

Finally, it is worth mentioning that the 2-(2-pyrazolin-1-yl)-thiazole scaffold was the most common and was responsible for the majority of the results concerning the antimicrobial activity of hybrid compounds containing thiazole and 2-pyrazoline heterocycles [16,17,18,19,20,21].

#### 2.2.2. Thiazolyl–Pyrazolin-3-One Hybrid Compounds

3-Pyrazolinone is an important scaffold for designing antimicrobial molecules [25], as is presented further.

Abu-Melha obtained some *N*-(4-pyrazolin-3-one)-2-thiazolyl-hydrazonomethyl-phenoxyacetamides, with various substituents in the fourth and fifth positions of the thiazole ring (Figure 16), through a multi-step synthesis between *N*-(4-antipyrinyl)-2-chloroacetamide, *p*-hydroxybenzaldehyde, thiosemicarbazide, and various α-halogenocarbonyl compounds [26].

The compounds were tested for their antibacterial activity against Gram-positive and Gram-negative strains and for their antifungal activity [26]. Compounds **66**–**68** showed superior antibacterial activity (MICs = 28–168 µg/mL) against *S. aureus* ATCC 25923, *Salmonella typhimurium* ATCC 14028, and *E. coli* ATCC 25922, compared to chloramphenicol (MICs = 143–152 µg/mL) and cephalothin (MICs = 135–229 µg/mL) [26].

In terms of antifungal activity, compounds **67** and **68** showed superior activity (MICs = 168–172 µg/mL) against *C. albicans* ATCC 10231, compared to cycloheximide (MIC = 254 µg/mL) [26].

Additionally, the compounds can target penicillin-binding proteins 4 (PBP4) from *S. aureus* and *E. coli* for antibacterial activity, based on molecular docking studies. The proposed target has not been verified in a biological assay [26].

#### 2.2.3. Thiazolyl–Pyrazole Hybrid Compounds

Pyrazole bears significant antimicrobial, anthelmintic, and anticancer properties, which makes this heterocycle an important motif when designing novel antimicrobial compounds [27]. Herein, we present the structure–activity relationship in thiazole clubbed with pyrazole compounds with promising antimicrobial potential, in order to establish how clubbing these two heterocycles influences biological activity.

Based on the structures found, it is possible to conclude that there were two types of scaffolds used: one in which the thiazole and pyrazole rings are clubbed through a linker, which is a methylylidenehydrazinyl, and the other, where both rings are linked directly. For the second type of scaffold, there were two possible linking positions to the pyrazole ring: one in the first position and the other in the third position of the pyrazole ring (Figure 17).

Gondru et al. [28] and Patil et al. [29] synthesized some series of 2-(pyrazol-4-yl)-methylylidenehydrazinyl-thiazoles (Figure 18). These compounds were substituted in the fourth position of a thiazole ring with various aryl and hetaryl substituents (R^1^), in the second position of a 2-pyrazoline ring with a benzothiazole or phenyl ring (R^2^), and in the fifth position with a coumarin (**69**–**74**) or a substituted phenyl ring (**75**–**86**) (R^3^) (Figure 18).

The compounds were assayed for their antibacterial activity against both Gram-positive and Gram-negative strains and for their antifungal activity against *Candida* sp. and *Aspergillus* sp. strains [28,29]. Compounds **69**, **70**, **72**, and **74** showed inferior activities (MICs = 1.9–7.8 µg/mL) against *S. aureus* MTCC 96 and MTCC 2940, *Micrococcus luteus* MTCC 2470, *K. planticola* MTCC 530, *E. coli* MTCC 739, and *P. aeruginosa* MTCC 2453, compared to ciprofloxacin (MIC = 0.9 µg/mL). Compounds **75**–**86** showed superior activities (MICs = 3.9–18.5 µg/mL) against *S. aureus*, *E. coli*, and *P. aeruginosa*, compared to chloramphenicol (MICs = 24.6–32.8 µg/mL) [28,29].

Concerning antifungal activity, compounds **69**, **71**, **72**, and **74** showed similar activities (MICs = 7.8 µg/mL) against *C. albicans* MTCC 3017 compared to miconazole (MIC = 7.8 µg/mL). Compounds **75**–**86** showed superior activities against *A. niger* and *C. albicans* (MICs = 3.9–11.3 µg/mL) [28,29].

Additionally, antibiofilm activity was evaluated on *S. aureus*, *K. planticola*, and *C. albicans* biofilms [28]. Compound **74** presented a promising biofilm inhibition against *S. aureus* MTCC 96 (IC_50_ = 11.8 ± 0.24 µM), while compound **72** inhibited the biofilm formation of *S. aureus* MLS16 MTCC 2940 (IC_50_ = 12 ± 0.14 µM), *K. planticola* MTCC 530 (IC_50_ = 14.07 ± 0.19 µM), and *C. albicans* MTCC 3017 (IC_50_ = 16 ± 0.11 µM) [28].

According to the structure–activity relationship study, inserting a strong electron-withdrawing group, particularly nitro, in the para position of the phenyl ring in R_1_ (**75**, **78**, **81**, and **84**) resulted in increased overall antibacterial and antifungal activity, while meta-substitution was not as favorable (**76**, **79**, **82**, and **85**) (Figure 18). Antibiofilm activity was favorably influenced by substitution with 8-bromocoumarin and benzo[*f*]coumarin heterocycles. The presence of benzothiazole moiety (**69**–**74**), which is known for its toxicity, could halt any future progress towards leader molecules [28,29].

Compounds **69**–**74** could target dehydrosqualene synthase of *S. aureus*, which is important for staphyloxanthin biosynthesis, a virulence factor. Based on molecular docking studies, the coumarin moiety is important for binding to Lys20 residue through hydrogen bonds. The proposed target has not been verified in a biological assay [28].

More antimicrobial 2-(pyrazol-4-yl)-methylylidenehydrazinyl-thiazole derivatives were reported by Matta et al. [30]. These compounds contain a 4-(1-phenyl-1,2,3-triazol-4-yl)-methoxyphenyl moiety in the third position of the pyrazole ring. The phenyl ring directly linked to the 1,2,3-triazole ring contains two points of variation (R^1^ and R^2^), while the third point of variation (R^3^) is on the phenyl ring linked to the fifth position of the thiazole ring (Figure 19) [30].

The compounds were tested for their antibacterial activity, using an MIC assay, against both Gram-positive and Gram-negative strains [30]. On the one hand, compounds **88**–**90** showed inferior activity against *S. aureus* MTCC 96 (MICs = 4.1 ± 0.03–5.1 ± 0.08 µg/mL) compared to novobiocin (MIC = 3.9 ± 0.03 µg/mL) and against *E. coli* MTCC 443 (MICs between 10.0 ± 0.06 and >25 µg/mL) compared to ampicillin (MIC = 3.9 ± 0.03 µg/mL). On the other hand, the same compounds showed superior activity against *P. aeruginosa* MTCC 424 (MICs = 8.7 ± 0.08–9.8 ± 0.03 µg/mL) compared to ampicillin (MIC = 10.0 ± 0.08 µg/mL) [30].

Regarding the antifungal activity, the compounds were tested against *A. niger* MTCC 404, using the disk diffusion method [30]. Compounds **87**, **89**, and **90** showed inferior activity against *A. niger* (IZs = 3.0 ± 0.11–7.3 ± 0.06 mm) compared to miconazole (IZ = 8.0 ± 0.11 mm), at 10 µM concentrations [30].

SAR studies of this series showed that bromo-substituted compounds (R^3^ = Br in **87**, **89**, and **90**) had the best antifungal activity against *A. niger* compared to all the tested compounds (Figure 19). Chloro substitution in the same position (R^3^ = Cl) increased the activity against *P. aeruginosa*, as seen in compound **88**, while dimethylation (R^1^ = R^2^ = CH_3_) increased the activity against *S. aureus*, as seen in compound **90** [30].

The inhibition of topoisomerase IV has been proposed as a target for antibacterial activity against *S. aureus*, based on molecular docking studies. Additionally, the compounds can potentially target the main protease (Mpro) of SARS-CoV-2 virus. None of the proposed targets have been verified in biological assays [30].

Abdel-Aziem et al. [31] and Kumar et al. [32] designed some 5-(coumarin-3-yl)-2-(pyrazol-1-yl)-thiazoles. These compounds were halo-substituted in the sixth position of the coumarin ring (R^1^) and variously substituted in the third and fourth positions of the pyrazole ring (R^2^ and R^3^) (Figure 20).

The compounds were evaluated for their antibacterial activity, using the agar well diffusion method or MIC assay, against both Gram-positive and Gram-negative strains and were evaluated for their antifungal activity against *Candida* sp. strains [31,32]. Compounds **91**–**94** showed superior activity against *Enterococcus faecalis* ATCC 29212, compared to chloramphenicol, while compounds **93** and **94** showed superior activity against *P. aeruginosa* ATCC 27853, compared to cephalotin [31]. Both compounds **95** and **96** showed superior activity (MICs = 15.67–31.25 µM) against *S. aureus* MTCC 3160, *S. pyogenes* MTCC 442, and *E. faecalis* MTCC 439, compared to kanamycin (MICs = 31.25–62.50 µM) [32].

Regarding antifungal activity, compounds **95** and **96** showed inferior activity (MIC = 61.25 µM) against *C. albicans* NCPF 400034, *C. keyfer* NCPF 410004, *C. krusei* NCPF 44002, and *C. parapsilosis* NCPF 450002, compared to amphotericin B (MICs = 0.78–12.50 µM) [32].

SAR studies showed that small substituents, like methyl (**91**) and hydroxy (**92**), were important for activity against *E. faecalis*, while the larger substituents, like trifluoromethyl (**93**), were important for activity against *P. aeruginosa* (Figure 20). Moreover, only the compounds with both R^1^ and R^3^ substituents being electron-withdrawing groups (**95** and **96**) displayed overall improved antibacterial and antifungal activity. No potential target was reported by the authors [31,32].

Mahmoodi and Ghodsi designed some 2-pyrazolium-thiazol-4-yl salts substituted in the fourth position of the thiazole ring with a coumarin, in the third position of a pyrazolium ring with various substituted coumarins (R^2^), and in the fifth position with various aryl and hetaryl substituents (R^1^) (Figure 21) [33].

The compounds were tested, using the zone inhibition method, for their antibacterial activity against both Gram-positive and Gram-negative strains and for their antifungal activity against *Aspergillus* sp. strains [33]. Compounds **97**–**103** and **105** showed inferior activity (IZs = 12–17 mm) against *S. aureus*, *E. coli* and *M. luteus* compared to gentamycin (IZs = 18–21 mm) [33].

Only compound **102** was active (IZs = 16–17 mm) against *A. niger* and *A. flavus* but was inferior to fluconazole (IZ = 25 mm) in terms of antifungal activity [33].

Nevertheless, the heterogeneity of the results and the lack of activity in MIC terms make it difficult to draw conclusions about potential structure–activity relationships. No potential target was reported by the authors [33].

Nalawade et al. designed a series of 2-phenyl-5-(4-hetaryl-pyrazol-3-yl)-thiazoles (Figure 22) [34].

The compounds were tested, using the well diffusion method, for antibacterial activity against Gram-positive and Gram-negative strains and for antifungal activity against three types of strains [34]. All of the tested compounds (**108**–**124**) showed inferior activity (IZs = 9.6–14.4 mm) against *E. coli* and *S. epidermidis*, compared to streptomycin (IZs = 18.52–25.0 mm). In terms of antifungal activity, all of the compounds were active (IZs = 13.0–22.3 mm) against *C. albicans* NCIM 3100, *A. niger* ATCC 504, and *Rhodotorula glutinis* NCIM 3168 but were inferior to fluconazole (IZs = 18.35–25.30 mm) and ravuconazole (IZs = 20.15–28.64 mm) [34].

Antifungal activity was further evaluated through MIC screening [34]. Eleven compounds (**108**–**118**) emerged as promising anti *A. niger* agents (MICs = 31.25 µg/mL), with similar activity compared to ravuconazole (MICs = 7.81–31.25 µg/mL). Twelve compounds (**109**, **111**–**115**, **119**–**124**) were moderately active against *R. glutinis* (MICs = 62.5 µg/mL), and only one (**110**) was active against *C. albicans* (MIC = 62.5 µg/mL) but with inferior activity compared to fluconazole (MIC = 7.81 µg/mL) and ravuconazole [34].

The structure–activity relationship in these compounds implies that the best activity against *A. niger* was in the presence of methyl (**110**–**112**) or fluoro (**113**–**117** and **120**), as substituents on the phenyl ring directly linked to thiazole, while bulkier substituents, such as chloro (**117**, **121**, and **122**) and bromo (**118**, **119**, and **122**–**124**), were associated with lower activity (Figure 22). This was the opposite in the case of anti *R. glutinis* activity [34].

The antifungal activity of these compounds can be attributed to their capacity to target lanosterol 14α-demethylase, based on molecular docking studies. The proposed target has not been verified in a biological assay. No potential target was reported by the authors for antibacterial activity [34].

To conclude the results observed in the papers analyzed (Figure 23), clubbing thiazole with pyrazole to obtain novel antimicrobials seems to expand the activity spectrum, compared to 2-pyrazoline. Pyrazole-containing compounds displayed antibacterial activity against Gram-positive and Gram-negative strains, while antifungal activity became increasingly better. However, it should be noted that additional structural elements, such as a hydrazine linker or a supplementary heterocycle, like coumarin or 1,2,3-triazole, could significantly influence antimicrobial activity. For example, coumarin-containing compounds were only active against Gram-positive bacterial and fungal strains, while 1,2,3-triazole-containing compounds were potent antifungal agents. Also, the importance of the hydrazine linker in designing antifungal compounds was highlighted in the literature [35,36].

Based on the results provided, the general structure–activity relationship studies of antimicrobial 3-(2-(pyrazol-1-yl)-thiazol-4-yl)-coumarins and 2-pyrazolium-thiazol-4-yl salts could be formulated as follows: a halogen atom, bromo or chloro, in the sixth position of the coumaryl moiety enhances, but is not essential to (as observed in the pyrazolium series, Figure 21), the antimicrobial potential, while the nature of the substituents from pyrazole or pyrazolium moieties influences the spectrum. Thus, compounds containing electron-withdrawing groups had a larger span of activity compared to those containing electron-donating groups, covering both Gram-positive and Gram-negative bacterial strains, as well as fungal strains.

Nevertheless, clubbing pyrazole with 1,3-thiazole is a promising research hypothesis when designing novel antifungals.

#### 2.2.4. Thiazolyl–Imidazole Hybrid Compounds

Imidazole is frequently present in various bioactive compounds. Besides the potent activity against fungal strains, imidazole is found in compounds with various effects, such as antibacterial, antituberculosis, antiviral, antiparasitic, or anticancer [37].

Nikalje et al. reported the synthesis of some 2-(2,4,5-triphenyl-imidazol-1-yl)-thiazoles, variably substituted in the second position of imidazole ring (Figure 24) [38]. The general structure of these series was rationally designed by using the thiazole heterocycle from abafungin, isavuconazole, and ravuconazole; the imidazole heterocycle; and the two phenyl rings from clotrimazole, flutrimazole, and bifonazole [38].

The compounds were tested for their antifungal activity against multiple strains. The activity was quantified using MIC_80_, which is the minimal inhibitory concentration for an 80% inhibition in growth [38]. Among the synthesized compounds, those containing either electron-donating groups (**125**–**128**) or nitro (**129**) on the aryl showed superior activity, in most cases, against *C. albicans* NCIM 3471 and *C. glabrata* NCYC 388 (MIC_80_ = 0.2–0.35 µg/mL), *Fusarium oxysporum* NCIM 1332 (MIC_80_ = 20–35 µg/mL), *A. flavus* NCIM 539 (MIC_80_ = 35–50 µg/mL), *A. niger* NCIM 1196 (MIC_80_ = 40–45 µg/mL), and *C. neoformans* NCIM 576 (MIC_80_ = 5–10 µg/mL), compared to miconazole and fluconazole (MIC_80_ between 0.5 µg/mL and >64 µg/mL) [38].

The antifungal activity of these compounds can be attributed to their capacity to target lanosterol 14α-demethylase, based on molecular docking studies. The proposed target has been confirmed through an ergosterol extraction and quantitation assay, and it was found that these compounds can inhibit ergosterol biosynthesis [38].

Dekate et al. reported a series of 2-(iminobenzylidene)-5-(imidazole-2-yl)-thiazole derivatives, variably substituted on the phenyl ring linked in the fourth position of the thiazole ring (R^1^) and on the iminobenzylidene moiety (R^2^) (Figure 25) [39].

The compounds were tested for their antibacterial activity against Gram-positive and Gram-negative strains, using ciprofloxacin as a reference [39]. Compounds **130** and **131** showed inferior activity against *E. coli* (IZs = 25–27 mm) compared to ciprofloxacin (IZ = 34 mm), while compound **130** showed equal activity to the reference against *S. aureus* (IZ = 32 mm) [39].

Regarding the antifungal activity, the compounds were tested against *C. albicans* and *A. niger* strains, using clotrimazole as reference [39]. Only compound **132** showed superior activity against *C. albicans* (IZ = 25 mm) compared to clotrimazole (IZ = 22 mm) [39].

SAR studies of this series showed that dichloro substitution (R^1^ = R^2^ = Cl) increased antifungal activity against *C. albicans*, as seen in compound **132** (Figure 25) [39]. The substitution of both points of variation with large electron-withdrawing substituents, such as bromo and nitro in **130**, increased antibacterial activity against *S. aureus*. Mono-substitution with an electron-withdrawing substituent (R^1^ = Cl in **131**) decreased the activity against *S. aureus* but slightly increased antibacterial activity against *E. coli*, compared to compound **130** (Figure 25). No potential target was reported by the authors [39].

#### 2.2.5. Thiazolyl–Thiazolidin-4-One Hybrid Compounds

Thiazolidin-4-one is another versatile five-membered heterocycle, used in designing novel antibacterial compounds—one important direction being the development of antituberculosis compounds [40].

Othman et al. designed a series of novel 2-(thiazol-2-yl)-*N*-thiazolidin-4-ones, variably substituted in the second position of the thiazolidin-4-one ring (Figure 26) [41].

The compounds were tested for their antibacterial activity against sensitive (ATCC 25177 H37Ra), MDR (multidrug resistant, ATCC 35822), and XDR (extended drug resistant, RCMB 2674) strains of *Mycobacterium tuberculosis* and various bronchitis-causing bacteria [41]. Five compounds (**133**–**137**) showed equal or inferior activity against the sensitive strain, compared to isoniazid (MIC = 0.12 µg/mL). Three compounds (**133**, **135**, and **137**) showed activity against the MDR strain (MICs = 1.95–7.81 µg/mL), and one (**135**) showed activity against the XDR strain (MIC = 7.81 µg/mL). Concerning antibacterial activity, all five compounds showed similar activity (MICs = 0.48–7.81 µg/mL) against *Mycoplasma pneumoniae* ATCC 15531, *S. pneumoniae* ATCC 1659, and *K. pneumoniae* ATCC 43816, while four compounds (**133** and **135**–**137**) showed similar activity against *Bordetella pertussis* ATCC 9340 (MICs = 1.95–7.81 µg/mL), compared to azithromycin (MICs = 0.49–7.81 µg/mL) [41].

SAR studies showed that the activity on the *M. tuberculosis*-sensitive strain was favorably influenced by the presence of halo-substituents in para position (**135** and **137**) or methoxy substituents (**133**, **134**, and **136**). Mono-substitution (**133**, **134**, and **136**) was associated with MDR antituberculosis activity. Supplementarily, the presence of a voluminous halogen in the fourth position (**135**) was associated with XDR antituberculosis activity, most likely due to increased molecular lipophilicity (Figure 26) [41].

Concerning the activity against bronchitis-causing bacteria, SAR studies showed that the best antibacterial activity was associated with the grafting of methoxy groups in the meta position (**134** and **136**) (Figure 26) [41].

Molecular docking studies showed that the compounds can target the enoyl-acyl carrier protein reductase InhA of *M. tuberculosis*, important for type II fatty acid biosynthesis. Inhibition is enhanced by the substituent from the fourth position of the thiazole, the carbonyl of the ester group binding to the target through an accepting hydrogen bond. The inhibitory activity on InhA was further confirmed by in vitro studies [41].

Abo-Ashur et al. reported the synthesis of a series of 2-(thiazol-2-yl)-imino-thiazolidin-4-ones (Figure 27) [42].

The compounds were tested for their antituberculosis activity, with six compounds (**140**–**145**) presenting similar activity (MICs = 0.78–3.12 µg/mL) to isoniazid (MIC = 0.78 µg/mL) against *M. tuberculosis* RCMB 010126 [42].

Antibacterial activity was also tested against Gram-positive and Gram-negative strains [42]. Three compounds (**142**, **143**, and **146**) showed excellent activity against *S. aureus* RCMB 010028 and *P. aeruginosa* RCMB 010043 (MICs = 0.49–0.98 µg/mL), and six compounds (**142**–**146**) showed excellent activity against *E. coli* RCMB 010052 (MICs = 0.49–0.98 µg/mL), compared to ciprofloxacin (MICs = 1.95–3.90 µg/mL) [42].

With respect to antifungal activity, which was tested against *Candida* and *Aspergillus* strains, two compounds showed superior activity (**142** and **146**) against *A. fumigatus* (MICs = 0.49 µg/mL), and four compounds (**139**, **142**, **143**, and **146**) showed superior activity against *C. albicans* (MICs = 0.49 µg/mL), compared to amphotericin B (MICs = 0.98–1.95 µg/mL) [42].

SAR studies of these series suggest that a halogen (**138** and **141**) or methoxy (**139**, **143**, and **144**) substituent grafted on the phenyl ring (R_2_) is essential for antituberculosis, antibacterial, and antifungal activities (Figure 27). Advantageous for these activities were also the bioisosteric substitution of the phenyl ring with 3-pyridinyl (**146**) and the grafting of an additional morpholinyl ring (**140** and **142**) [42]. No potential target was reported by the authors.

#### 2.2.6. Thiazolyl–Thiazolidindione Hybrid Compounds

Widely known for its antidiabetic activity in glitazones, the thiazolidinedione heterocycle was also used in designing new antibacterial, antifungal, antiretroviral, antituberculosis, and anticancer compounds [43].

Alegaon et al. designed a series of antimicrobial 2-(thiazolidin-2,4-dion-3-yl)-*N*-(thiazol-2-yl) acetamides (Figure 28). The active methylene from the fifth position of the thiazolidin-2,4-dione ring was derivatized (R) with various aromatic and heteroaromatic aldehydes through Knoevenagel condensations [44].

The compounds were tested for their antibacterial activity against Gram-positive and Gram-negative strains and for their antifungal activity against various fungal strains [44]. Five compounds (**147**–**151**) showed inferior activity (MICs = 4–32 µg/mL) against *S. aureus* ATCC 25923, *E. faecalis* ATCC 35550, *E. coli* ATCC 35218, and *P. aeruginosa* ATCC 25619, compared to ciprofloxacin (MICs = 2 µg/mL) [44].

Concerning antifungal activity, the same compounds showed inferior activity against *C. albicans* ATCC 2091, *A. flavus* NCIM 524, *A. niger* ATCC 6275, and *C. neoformans* (clinical isolate), compared to ketoconazole (MICs = 1–2 µg/mL) [44].

SAR studies underlined the finding that the substitution of the arylidene moiety from the fifth position of the thiazolidin-2,4-dione ring with electron-donating groups, especially methoxy (**148**–**150**), enhanced the overall antibacterial and antifungal activities (Figure 28). A bioisosteric substitution of the phenyl ring with 2-furanyl (**151**) also favorized antibacterial and antifungal activities. Substitution with electron-withdrawing groups, such as nitro or cyano, almost abolished both activities [44].

These compounds could potentially target the ATP binding domain of bacterial DNA gyrase B and the fungal lanosterol 14α-demethylase, according to molecular docking studies. The proposed targets were confirmed by studying the binding interactions of these compounds with circulating tumor DNA (ct-DNA), using absorption spectroscopy [44].

#### 2.2.7. Thiazolyl–1,3,4-Thiadiazole Hybrid Compounds

The versatility of 1,3,4-thiadiazole heterocycle comes from its ability to act as a hydrogen-binding domain, as a two-electron donor system, and as a bioisosteric replacement of thiazole. Thus, it is a valuable moiety for designing novel antimicrobial, anticancer, and antiprotozoal compounds [45].

Using artificial intelligence, Stokes et al. repurposed an antidiabetic compound, SU-3327, which acts as a c-Jun N-terminal protein kinase (JNK) inhibitor, into a promising antibacterial molecule called halicin. Structurally, halicin (**152**) contains 5-nitrothiazole and 2-amino-1,3,4-thiadiazole, linked in the 2,5′ positions by a thioether group (Figure 29) [46,47].

Halicin displayed broad antibacterial activity against *E. coli* (MIC = 2 µg/mL), *M. tuberculosis*, carbapenem-resistant Enterobacteriaceae, and *A. baumanii* but no activity against *P. aeruginosa* [46]. Further studies for the assessment of its antibacterial potential were reported by other authors as well [47,48].

Booq et al. tested halicin with significant results against *S. aureus* ATCC BAA-977 (MIC = 16 µg/mL), *E. coli* ATCC 25922 (MIC = 32 µg/mL), *A. baumanii* ATCC BAA-747 (MIC = 128 µg/mL), and *A. baumanii* MDR isolate (MIC = 256 µg/mL) bacterial strains [47]. Hussain et al. obtained promising results when halicin was assayed against strains of *E. faecalis* and *E. faecium* (MICs = 4–8 µg/mL) [48]. Moreover, halicin showed proof of antibiofilm potential, not only alone but also in synergism with other molecules, as reported by some authors [49,50]. Currently, there are no ongoing clinical trials for testing halicin on humans.

Its unique mechanism of action, which implies the disruption of a membrane electrochemical gradient, pH modification, and the upregulation of iron acquisition genes, makes it difficult to acquire resistance. It is also possible that halicin may act as a siderophore prior to pH alteration [46].

#### 2.2.8. Thiazolyl–1,2,3-Triazole Hybrid Compounds

1,2,3-Triazole is the most stable among heterocycles with three adjacent nitrogen atoms, and it can be found in variable bioactive compounds, including antibacterial, antifungal, antimalarial, and anticancer agents [51]. Herein, we present a structure–activity relationship in thiazole clubbed with 1,2,3-triazole compounds, with promising antimicrobial potential, to establish how clubbing these two heterocycles could potentially influence biological activity.

Based on the structures found, it is possible to conclude that there were two types of scaffolds used: one in which the thiazole and 1,2,3-triazole rings are linked directly but with different linking positions (**a**–**b**) [52,53], and the other one in which the rings are clubbed through various linkers and variable linking positions (**c**–**e**) [54,55,56,57] (Figure 30).

Shinde et al. reported the design of novel antituberculosis 5-(1,2,3-triazol-4-yl)-thiazoles, substituted in the second position of the thiazole ring with various aryl substituents (Figure 31) [52].

Antituberculosis activity was tested against *M. tuberculosis* H37Ra (ATCC 25177), using rifampicin (IC_50_ = 0.002 µg/mL and MIC_90_ = 0.75 µg/mL) as reference. The activity of the compounds was quantified using both IC_50_ and MIC_90_. While most of the compounds were very active in terms of IC_50_ values (0.58–8.23 µg/mL) [52], only two compounds (**153** and **154**) were active in terms of MIC_90_ values (2.22 µg/mL and 4.71 µg/mL), with the activity being inferior to rifampicin [52].

SAR studies show that a fluoro substitution of the benzyl group linked to 1,2,3-triazole is responsible for the antituberculosis activity (Figure 31) [52]. However, this boost in the activity only took place when the other phenyl ring was either unsubstituted (**153**) or 3-methyl-substituted (**154**). Double halogen substitution was associated with a decrease in the activity [52].

These compounds can potentially target enoyl-acyl carrier protein reductase, which is an important enzyme in the fatty acid biosynthesis and growth of mycobacteria, based on molecular docking studies. The proposed target has not been verified in a biological assay [52].

Mahale et al. synthesized a series of 2-(1,2,3-triazol-4-yl)-thiazole derivatives, substituted on the first position of the 1,2,3-triazole ring (R^1^) and the fifth position of the thiazole ring (R^2^) with aryl substituents (Figure 32) [53].

The compounds were tested for their antibacterial activity against Gram-positive and Gram-negative strains and for their antifungal activity against *Candida* sp. and *Aspergillus* sp. strains [53]. Compounds **156**, **158**, **160**, and **162**–**166** were equally active (MIC = 0.5 µg/mL) against *E. coli* ATCC 25922 and *S. aureus* ATCC 25923, compared to streptomycin (MIC = 0.5 µg/mL).

Concerning antifungal activity, compounds **155**–**157**, **159**, **161**, and **162** were equipotent (MIC = 0.5 µg/mL) against *C. albicans* MTCC 2977 and *A. niger* MCIM 545, compared to griseofulvin (MIC = 0.5 µg/mL) [53].

SAR studies suggest that substitution with predominantly electron-withdrawing groups, like nitro and halogen atoms, provide an antibacterial effect to the molecules, while an antifungal effect can be achieved using electron-donating, such as methyl and methoxy, and electron-withdrawing groups in both positions (Figure 32). No potential target was reported by the authors [53].

Jagadale et al. designed a series of 1-(thiazol-5-yl)-2-(1,2,3-triazol-1-yl)-ethanol derivatives, substituted in the second position of the thiazole ring (R^1^) and the fourth position of the 1,2,3-triazole ring (R^2^) with various aryl substituents (Figure 33) [54].

The compounds were tested for their antibacterial activity against Gram-positive and Gram-negative strains and for their antifungal activity against various strains [54]. Compounds **168**–**170** showed inferior activity (MIC = 62.5 µg/mL) against *S. epidermidis* NCIM 2178, compared to streptomycin (MIC = 7.81 µg/mL) [54].

Regarding antifungal activity, compounds **167** and **171**–**174** were inferior against *A. niger* (ATCC 504) compared to fluconazole (MIC = 7.81 µg/mL) but similar to ravuconazole (MIC = 31.25 µg/mL). Additionally, compound **170** displayed activity against *R. glutinis* (MIC = 62.5 µg/mL) too but displayed inferior activity to both reference compounds [54].

SAR studies suggest that substitution with 4-chloro (**172**–**174**) and 4-fluoro (**171**) in R^1^, as well as unsubstituted phenyl (**167**), are responsible for activity against *A. niger* (Figure 33). A fluoro substitution of only one (**168**–**169**, and **172**) or both (**170**) of the phenyl rings was associated with antibacterial activity against *S. epidermidis*. Double 4-fluorophenyl substitution induced activity against *S. epidermidis*, *A. niger*, and *R. glutinis*, as seen in compound **170**. No potential target was reported by the authors [54].

Poonia et al. designed some 2-(1,2,3-triazol-1-yl)-*N*-(thiazol-2-yl)-acetamides and 2-(1,2,3-triazol-1-yl)-*N*-(benzothiazol-2-yl)-acetamides, para-substituted on the phenyl-ureidomethyl moiety, linked to the fourth position of the 1,2,3-triazole ring (Figure 34) [55].

The compounds were tested for their antibacterial activity against Gram-positive and Gram-negative strains and for their antifungal activity against *Candida* sp. and *Rhizopus* sp. strains [55]. All of the compounds (**175**–**186**) showed superior activity (MICs = 0.0074–0.0333 µmol/mL) against *C. albicans* MTCC 183 and *Rhizopus oryzae* MTCC, compared to fluconazole (MIC = 0.0408 µmol/mL) [55].

Concerning antibacterial activity, four compounds (**177**, **182**, **185**, and **186**) showed noteworthy activity. Compounds **177** and **182** showed inferior activity (MICs = 0.0287–0.0299 µmol/mL) against *E. coli* MTCC 1654 and *S. aureus* MTCC 3160, compared to ciprofloxacin (MIC = 0.0094 µmol/mL). Compounds **182** and **185** showed inferior activity (MICs = 0.0257–0.0299 µmol/mL) against *P. fluorescens* MTCC 664, compared to the reference drug, while compound **186** showed superior activity (MIC = 0.0071 µmol/mL). Lastly, compound **183** was the most active against *Bacillus endophyticus* (MIC = 0.0257 µmol/mL) but still inferior to the reference drug [55].

Based on SAR studies, the insertion of thiazole or benzothiazole rings increased antifungal activity, compared to the phenylureidopropargyl precursors and fluconazole (Figure 34) [55]. For antibacterial activity, bromo (**177**, **181**, and **185**) and methoxy (**178**, **182**, and **183**) substitutions, as well as the annulation of the thiazole ring (**183**–**186**), were the most important [55].

These compounds act as antifungals by targeting sterol 14-α demethylase, according to molecular docking studies. The proposed target has not been verified in a biological assay [55].

More 2-(1,2,3-triazol-1-yl)-*N*-(thiazol-2-yl)-acetamides were reported by Veeranki et al. [56]. These compounds contained a *N*-methylpiperazine moiety linked to the fourth position of the 1,2,3-triazole ring. This moiety was substituted with various alkyl and aryl substituents on the other nitrogen atom (Figure 35) [56].

The compounds were tested for their antimicrobial activity against bacterial and fungal strains [56]. Compounds **188**–**190** showed inferior activity (IZs = 14–15 mm) against *E. coli*, *P. aeruginosa*, and *S. aureus*, compared to ampicillin (IZs = 25–24 mm) [56].

Their antifungal activity was tested against *C. albicans* and *A. niger*, using clotrimazole as reference. Compounds **187**–**190** showed the best activity against *C. albicans* (IZs = 15–18 mm), compared to all the tested compounds [56].

Based on the SAR studies of this series, 4-fluoro substitution induced an overall favorable antimicrobial activity, while 3-chloro and pentyl substitutions selectively increased antifungal activity against *C. albicans* (Figure 35) [56]. Additionally, substitution with ethyl hexanoate was favorable for antibacterial activity [56].

These compounds act as antibacterials by potentially targeting glucosamine-6-phosphate synthase (GlmS), important in microbial cell membrane synthesis, and as antifungals by targeting candidapepsin-1, important in candidiasis infections, based on molecular docking studies. None of the proposed targets have been verified in biological assays [56].

Gondru et al. reported the synthesis of two series of 2-(1,2,3-triazol-4-yl-methoxybenzylidenehydrazinyl)-thiazoles, substituted on the fourth position of thiazole ring with aryl and hetaryl substituents (Figure 36) [57]. They used para-hydroxybenzaldehyde as starting material, of which the phenolic hydroxyl group was etherified with a substituted 1,2,3-triazole moiety, while the carbonylic group was derivatized to corresponding hydrazonothiazoles [57].

The compounds were tested for their antibacterial activity against Gram-positive and Gram-negative strains and for their antifungal activity against various *Candida* sp. and *Issatchenkia* sp. strains [57]. Compounds **191**–**197** showed inferior activity (MICs = 2.8–15.7 µM) against *S. aureus* MTCC-96, *S. aureus* MLS16 (MTCC 2940), *M. luteus* MTCC 2470, *K. planticola* MTCC 530, *E. coli* MTCC 739, and *P. aeruginosa* MTCC 2453, compared to ciprofloxacin (MIC = 2.7 µM) [57].

Concerning antifungal activity, compounds **192**–**196** showed superior activity (MICs = 5.9–14.2 µM) against *C. albicans* MTCC 183, MTCC 854, and MTCC 3018, *C. aaseri* MTCC 1962, *C. glabrata* MTCC 3019, and *Issatchenkia hanoiensis* MTCC 4755, compared to miconazole (MIC = 18.7 µM) [57].

According to SAR studies, 4-methoxyphenyl (**191**), benzo[f]coumarinyl (**192**), 8-methoxycoumarinyl (**193**), 6-bromo-8-methoxycoumarinyl (**194**), and 8-bromocoumarinyl (**195**) substitutions were associated with antibacterial activity against *S. aureus* (Figure 36) [57]. The presence of the coumarin heterocycle was important for antifungal activity, especially against the *C. albicans* MTCC 183 strain. By replacing the methoxy group from the coumarin ring with an ethoxy group or introducing electron-withdrawing groups, like chloro and nitro, in both the 6- and 8- positions of the coumarin, the antibacterial effect was canceled but not the antifungal one. No potential target was reported by the authors [57].

In conclusion, 1,2,3-triazole heterocycle is a versatile moiety for designing novel antimicrobial compounds with a broad activity spectrum. As observed in the studies presented, clubbing with thiazole resulted in potent compounds against a large variety of pathogen strains, including mycobacteria.

A prominent feature of the presented compounds was the presence of substituted phenyl rings, linked directly to the heterocycles or through a linker. Thus, the differences between compound activities were mostly attributed to these substituents. By far, the most used substituents in these compounds were halogens as well as methyl and methoxy groups [52,53,54,55,56,57]. A summary of the structure–activity relationships in antimicrobial 1,3-thiazole clubbed with 1,2,3-triazole hybrid compounds is presented in Figure 37.

#### 2.2.9. Thiazolyl–1,3,4-Oxadiazole Hybrid Compounds

Similar to the previously mentioned heterocycles containing three heteroatoms, the 1,3,4-oxadiazole ring can be found in a plenitude of compounds with various biological activities, including anticancer, antibacterial, antifungal, and antiviral effects. This heterocycle can act as a bioisostere for the carbonyl group and can be used in the structure of a molecule as a flat aromatic linker to ensure an adequate orientation [58].

As reported in the literature, the 5-thioxo-1,3,4-oxadiazole heterocycle can be found in a series of anticancer and antimicrobial compounds [59,60,61], thus making this heterocycle an important contender for designing novel antimicrobials.

Some series of 2-thiazolyl-5-mercapto-1,3,4-oxadiazoles, either linked through a methylene linker between the fourth position of the thiazole ring and the second position of the 1,3,4-oxadiazole ring, or directly linked between the fifth position of the thiazole ring and the second position of the 1,3,4-oxadiazole ring, were synthesized [62,63].

These compounds exist in two tautomeric forms (Figure 38). Existing data [64] show that the thione tautomer (**II**) is more stable in the solid state, while, in a solution, the thiol tautomer (**I**) is predominant. The existence of thiol–thione tautomerism was valorized by obtaining the corresponding thioether derivatives and Mannich bases [62,63].

Athar Abbasi et al. described the synthesis of 2-(thiazol-4-yl)-5-thio-1,3,4-oxadiazoles, capable of urease inhibition, thus offering an alternative potential treatment to *Helicobacter pylori* infections (Figure 39) [62].

Their inhibitory activity was tested using thiourea (IC_50_ = 21.11 ± 0.12 µM) as a reference. Based on the results (IC_50_ = 2.17 ± 0.41 µM) and molecular docking studies, compound **198** (Figure 39) presented the best binding affinity (-8.40 kcal/mol) among all the synthesized and tested compounds, and was able to bind to the active site of the enzyme [62].

By integrating all the information obtained, it is possible to conclude that the fluorine atom forms two halogen–metal bonds with the active urease nickel center. The potential inhibition of urease was influenced by the type and position of the halogen atom on the benzyl moiety linked to the sulfur atom. The fluorine atom in para position (**198**) was the most advantageous for this inhibition. For chlorine, the meta position was favorable, while for bromine, either the ortho or para position was favorable. One more important aspect observed was that an unsubstituted benzyl moiety yielded the weakest inhibitory capacity; therefore, the presence of a halogen substituent was essential for urease inhibition [62].

Desai et al. designed a series of 2-(thiazol-5-yl)-5-mercapto-1,3,4-oxadiazole Mannich bases. These compounds are variably substituted on the anilino moiety linked to the fourth position of the 1,3,4-oxadiazole ring, through a methylene bridge (Figure 40) [63].

The compounds were tested for their antibacterial activity against Gram-positive and Gram-negative bacterial strains and for their antifungal activity against *Candida* sp. and *Aspergillus* sp. strains [63]. Compounds **199**, **200**, and **203** showed superior activity against *E. coli* MTCC 443, *P. aeruginosa* MTCC 1688, and *S. pyogenes* MTCC 442, compared to ampicillin (MICs = 100–250 µg/mL) [63].

Concerning antifungal activity, compounds **201** and **202** showed superior activity (MICs = 25–50 µg/mL) against *C. albicans* (MTCC 227) and *A. niger* (MTCC 282), compared to griseofulvin (MICs = 100–500 µg/mL) [63].

SAR studies implied that the most active antibacterial compounds had electron-withdrawing groups, such as nitro (**203**) and fluoro (**199**–**200**), while the most active antifungal compounds had electron-donating groups, such as methoxy (**201**–**202**) (Figure 40). No potential target was reported by the authors [63].

### 2.3. Thiazole Clubbed with Five-Membered Benzofused Heterocycles

A benzofusion of heterocycles leads to an increase in lipophilicity and an electron-withdrawing character, which could be advantageous in drug design. Additionally, benzofused heterocycles constitute bioisostere alternatives to naphthalene, which can facilitate better binding to a potential target [65].

Based on the literature search of the last seven years, we will further discuss the structure–activity relationship in antimicrobial hybrid compounds containing thiazole clubbed with the following five-membered heterocycles and their derivatives: indole, carbazole, indolin-2-one, and tetrahydroindenofuran.

#### 2.3.1. Thiazolyl–Indole and Thiazolyl–Carbazole Hybrid Compounds

The antimicrobial properties of carbazole derivatives were known since the isolation of carbazole alkaloids from *Murraya koenigii* leaves [66]. Similarly, indole naturally occurs in various alkaloids with antimicrobial properties [67,68,69]. Since then, these heterocycles have been increasingly used in the design of synthetic novel antimicrobials.

Zhao et al. reported the design of a novel series of antimicrobial thiazole clubbed with either indole or carbazole hybrid compounds, containing a ferrocene scaffold (Figure 41) [70].

Antibacterial activity was assayed against *S. aureus*, *E. coli*, and *P. aeruginosa*, using ciprofloxacin as a reference (MIC = 15.625 µg/mL). Only three compounds (**204**–**206**) showed good activity against *E. coli* (MIC = 31.25 µg/mL), while the activities against the other strains were poor (MICs = 125–250 µg/mL) [70].

SAR studies of these series suggest that the introduction of a second heterocycle on the 4-ferrocenylthiazole scaffold, indole, or carbazole, significantly decreased antibacterial activity (Figure 41), compared to the 4-ferrocenyl-2-*N-*anilinothiazole series. No potential target was reported by the authors [70].

#### 2.3.2. Thiazolyl–Indolin-2-One Hybrid Compounds

Indolin-2-one derivatives are ubiquitous compounds found in the human body. They possess various biological activities, including antioxidant, antimicrobial, and antiproliferative activities [71].

Starting from 5-(piperazin-1-yl)-sulfonylindolin-2,3-dione, Alzahrani et al. obtained some clubbed thiazole Schiff bases and thiazolin-4-one compounds as dihydrofolate reductase (DHFR) inhibitors and anti-quorum sensing (QS) agents (Figure 42) [72].

They derivatized the carbonyl group from the third position of indolin-2,3-dione either by condensation with 2-*N*-anilino-thiazolin-4-one or with different aminothiazoles and hydrazonothiazoles (Figure 42, **207**–**210**) [72].

The compounds were tested for their antibacterial activity against Gram-positive and Gram-negative bacterial strains and for their antifungal activity against *C. albicans* ATCC 10213 [72]. Compounds **207**, **209**, and **210** showed superior activity (MICs = 1.9–15.6 µg/mL) against *S. aureus* ATCC 25923 and ATCC 29213, *E. faecalis* ATCC 29212, *E. coli* ATCC 35218, *P. aeruginosa* ATCC 9027 and PAR.36, and *S. typhimurium* ATCC 14028, compared to levofloxacin (MICs = 8.1–130 µg/mL). Furthermore, compounds **207**, **208**, and **210** inhibited the *E. faecalis* QS system at low concentrations and presented IC_50_ values of DHFR inhibition in the submicromolar range (0.04–0.28 µM), which were similar to methotrexate (IC_50_ = 0.061 µM) [72].

Regarding antifungal activity, as was expected, the compounds had inferior activity against *C. albicans* (**208**, MIC = 62.5 µg/mL), compared to nystatin (MIC = 3.9 µg/mL) [72].

SAR studies of these series can be constructed on four levels. The first level is the nature of the bond between the indolin-2-one ring and the thiazolin-4-one- or thiazole-containing fragments (Figure 42). Compound **207**, containing an ethenyl bond (C=C), showed antibacterial activity against *S. aureus* and *P. aeruginosa* (MIC = 12.5 µg/mL), while compounds **208**–**210**, containing an imine bond (C=N), showed both antibacterial and antifungal activities against a broader spectrum, including *E. faecalis*, *E. coli*, *S. typhimurium*, and *C. albicans* [72].

The second level is based on the nature of the azole heterocycle. As observed in the previous level, the compound containing a thiazolin-4-one ring had a narrower activity spectrum compared to the compounds containing the thiazole ring (Figure 42) [72].

The third level is based on the distance between the indolin-2-one ring and the thiazole ring. As the distance between the two rings grows, so does the antibacterial activity. Compound **208**, in which the rings are directly linked through an imine bond, had the weakest activity among all three imino compounds (MIC = 62.5 µg/mL). Compound **209** registered better activity when the imino bond was swapped with a hydrazono one (MIC = 7.8 µg/mL), while introducing a benzensulfonamide moiety in compound **210** resulted in the best antibacterial activity (MICs = 1.9–7.8 µg/mL). An additional sulfonamide group would increase antibacterial activity, which is the last level of the SAR studies of these series (Figure 42). The hydroxy group was able to increase antibacterial activity but decreased anti-quorum sensing activity [72].

It should be noted that replacing the methyl from the piperazine ring with phenyl decreased antibacterial activity (Figure 42) [72].

#### 2.3.3. Thiazolyl–Tetrahydroindenofuran Hybrid Compounds

Indenofurans are a rare occurrence in nature compounds, found in *Anisodus tanguticus* species. These compounds are mostly known for their antioxidant properties, but there have been reports about their antibacterial potential [73,74].

Adole et al. designed a series of 2-(2-tetrahydroindenofuranylidene)-hydrazinylthiazoles, substituted in the fourth position of the thiazole ring with various aryl groups (Figure 43) [75].

The compounds were tested for their antibacterial activity against Gram-negative strains and for their antifungal activity against various strains [75]. Compounds **211**–**215** showed inferior activity (MICs = 15.62–31.25 µg/mL) against *E. coli* and *Shigella boydii*, compared to ampicillin (MICs = 1.95–3.9 µg/mL).

Concerning antifungal activity, compounds **211**–**215** showed similar activity (MICs = 1.95–15.62 µg/mL) against *R. oryzae*, *Mucor mucedo*, *A. niger*, and *C. albicans*, compared to fluconazole (MICs = 0.97–1.95 µg/mL) [75].

SAR studies of these compounds underline the finding that para-substitution with bulky substituents was associated with overall good antibacterial and antifungal activities (Figure 43). Meta-substitution may be useful in enhancing the antifungal effect, with compound **214** being twice or thrice more active than the para-substituted compounds. It is also worth mentioning that fluoro substitution (**215**) is beneficial for antifungal activity but not for antibacterial activity [75]. Switching from phenyl to naphthyl in the fourth position of the thiazole ring decreased the overall antibacterial and antifungal activities, and this should be avoided. No potential target was reported by the authors [75].

### 2.4. Thiazole Clubbed with Six-Membered Heterocycles

Six-membered heterocyclic compounds are numerous and versatile in terms of structure and bioactivity. Based on the literature research in recent years, only thiazole clubbed with pyridine antimicrobials were reported [76,77,78,79]. Thus, we present SAR studies only concerning them.

#### Thiazolyl–Pyridine Hybrid Compounds

Pyridine is a privileged heterocycle found in a large variety of antimicrobial compounds, either as a sole heterocycle or together with other heterocycles. There have been reports that noted an additional heterocycle to pyridine may have the potential to enhance its biological activity [80].

Muluk et al. and Patil et al. designed some 2-(4-pyridyl)-thiazoles as potential DNA gyrase and lumazine synthase inhibitors [76,77,78]. These compounds are substituted in the fourth position of the thiazole ring with aryl or hetaryl substituents, linked to the thiazole ring via an acylhydrazonomethylene or α,β-unsaturated carbonyl linker (R^2^) (Figure 44). There is a supplementary substituent in the second position of the pyridine ring, which is ethyl or propyl, depending on the starting thiocarbamide fragment (CN=S) for the Hantzsch synthesis, ethionamide or prothionamide [76,77,78].

The compounds were tested for their antibacterial activity against Gram-positive and Gram-negative strains and for their antifungal activity against *Candida* sp. and *Aspergillus* sp. strains [76,77,78]. Compounds **216**–**220**, **224**, and **225** showed inferior activity (MICs = 18–170 µg/mL) against *S. typhi* ATCC 9207, *E. coli* ATCC 8739, *E. aerogenes* ATCC 13048, *S. aureus* ATCC 6538, and *P. aeruginosa*, compared to tetracycline (MICs = 3.0–20 µg/mL) and streptomycin (MIC = 30 µg/mL) [76,77,78].

Concerning antifungal activity, the same compounds showed inferior activity (MICs = 60–280 µg/mL) against *C. albicans* ATCC 10231, compared to fluconazole (MIC = 30 µg/mL) [76,77,78].

SAR showed that, by the bioisosteric substitution of the phenyl ring with a furan heterocycle (**220** and **224**) (Figure 44), antibacterial and antifungal activities increased. Substitution with electron-withdrawing groups was favorable for antibacterial and antifungal activities overall. The presence of a 2-nitro group (**216**) was favorable for activity against *E. aerogenes*, and 3,5-dichloro substitution (**219**) was favorable for activity against *C. albicans*—this being in agreement with the presence of this substituent in some important antifungal azoles like miconazole, ketoconazole, or itraconazole. The elimination of the methyl group from the fifth position of the thiazole ring (R^3^) did not affect the antibacterial and antifungal activities overall [76,77,78].

In order to establish the importance of the position by which the pyridine ring is linked to the thiazole one, Eryılmaz et al. synthesized two series of 2-(2/4-pyridyl)-thiazoles, substituted in the fourth position of the thiazole ring with aryl or hetaryl substituents (Figure 44) [79].

The compounds were tested for their antibacterial activity against Gram-positive and Gram-negative strains and for their antifungal activity against *C. albicans* ATCC 10231 [79]. Both 2-pyridyl (**229**–**233a**) and 4-pyridyl (**229**–**233b**) series showed inferior activity (MICs = 0.01–5.7 mM) against *E. coli* W3110, *P. aeruginosa* ATCC 27853, and *S. aureus* ATCC 6538P, compared to cefepime (MICs = 0.001–0.06 mM) and amikacin (MICs = 0.01–0.02 mM). The 4-pyridyl series (**b**) showed better activity (MICs = 0.01–4.7 mM) compared to 2-pyridyl series (**a**) (MICs = 1.2–5.7 mM) [79].

Concerning antifungal activity, both series showed superior activity against *C. albicans* compared to fluconazole; the 4-pyridyl series showed better activity (MICs = 0.15–1.2 mM) than the 2-pyridyl series (MICs = 0.15–2.8 mM) [79].

Based on these results, 4-pyridyl had a better influence on antibacterial and antifungal activities compared to 2-pyridyl. This could have been due to the better capacity of the compounds to interact with a potential target through hydrogen bonds or ionic bonds, which may be difficult in 2-pyridyl series due to steric hinderances [79].

Some of the compounds could express their antibacterial activity by targeting DNA gyrase, important for the replication of genetic material in bacteria, while some of the compounds expressed antifungal activity by targeting lumazine synthase, a key enzyme in the biosynthesis of riboflavin in fungi. The proposed targets have not been verified in a biological assay [76,77].

In conclusion, clubbing thiazole with pyridine to design potent antimicrobials in the cases presented mostly depends on the nature of the substituents used, which were mainly electron-withdrawing substituents (nitro and halogens). Based on the results obtained, the spectrum is narrower, and the activity is lower compared to hybrid compounds containing five-membered heterocycles, such as pyrazole or triazole. Thus, clubbing an additional heterocycle may be helpful; however, there was no similar study performed in the selected timeframe for this paper to support our hypothesis.

### 2.5. Thiazole Clubbed with Six-Membered Benzofused Heterocycles

Based on the literature search of the last seven years, we will further discuss the structure–activity relationship in antimicrobial hybrid compounds containing thiazole clubbed with the following six-membered heterocycles and their derivatives: coumarin, flavone, quinoline, quinoline-2(1*H*)-one, and quinazolin-4(3*H*)-one.

#### 2.5.1. Thiazolyl–Coumarin Hybrid Compounds

The coumarin heterocycle is found in some authorized antimicrobials (novobiocin and chlorobiocin) [81], thus making this moiety adequate for designing novel antimicrobials. Herein, we present the SAR studies of antimicrobial thiazole clubbed with coumarin compounds.

Yusufzai et al. reported the design and synthesis of some bis-coumarin derivatives (series **I**). In this series, the 4-(3-coumaryl)-thiazoles are linked to a second coumarin heterocycle by a methylylidenhydrazinyl bridge in the second position of the thiazole ring (Figure 45) [82].

The compounds were tested for their antibacterial activity against Gram-positive and Gram-negative strains and for their antituberculosis activity against the H37Rv strain (ATCC 25618) [82]. Compounds **234**–**236**, **238**, **240**, **242**, and **243** showed similar activities (MICs = 31.25–62.5 µg/mL) against *E. coli*, *E. aerogenes*, *S. typhi*, *S. pneumoniae*, and *S. aureus*, compared to streptomycin (MICs = 31.25–62.5 µg/mL), kanamycin (MICs = 62.5–125 µg/mL), and vancomycin (MICs = 31.25–250 µg/mL) [82].

Regarding antituberculosis activity, compounds **234**–**236**, **238**–**241**, **243**, and **245** showed inferior activity (MIC = 50 µg/mL) compared to isoniazid (MIC = 0.0781 µg/mL) [82].

SAR studies of these compounds suggest that the substitution of the second coumarin unit (R^2^) with a nitro group (**237** and **242**) abolished antituberculosis activity and severely decreased antibacterial activity (Figure 45) [82]. Bromo (**243**) and methoxy (**234**–**236** and **239**–**241**) substitutions on the same coumarin were advantageous for both activities, while hydroxy substitution (**244**) was by far the best for activity against all of the tested strains (MICs = 31.25–62.5 µg/mL). Coumarin heterocycles might have also made a significant contribution to the activity, as all of the compounds displayed different degrees of potency, but none of them were completely inactive. No potential target was reported by the authors [82].

Another attempt to obtain urease inhibitors against *H. pylori* was reported by Salar et al., who designed a series of 4-(3-coumaryl)-thiazoles, in which they replaced the second coumarin heterocycle with an aromatic structure (series **II**) (Figure 45) [83].

The inhibitory capacity of the compounds was compared to acetohydroxamic acid (IC_50_ = 27 ± 0.5 µM) [83]. The most active compound (**245**) had a better inhibitory activity than acetohydroxamic acid (IC_50_ = 16.29 ± 1.1 µM). Other results worth mentioning were registered by compounds **246** (IC_50_ = 76.41 ± 0.1 µM), **247** (IC_50_ = 77.67 ± 1.5 µM), and **248** (IC_50_ = 71.21 ± 1.6 µM) [83].

The structure–activity relationship in this series reflects the importance of fluoro, chloro, hydroxy, and methoxy substitutions for urease inhibition, as well as their relative position on the aryl ring (Figure 3). Thus, 2-fluoro (**247**), 2-hydroxy (**248**), 2-hydroxy-5-fluoro (**245**), and 2-chloro-3-methoxy (**246**) substitutions were the most advantageous for the inhibition [83].

Molecular docking studies further emphasized the capacity of these compounds to inhibit urease by binding to its active site [83].

Hu et al. designed two series of antibacterial 2-(8-(1-oxoprop-2-en-2-yl)-oxycoumarinyl)-thiazoles, substituted on the α,β-unsaturated carbonyl linker with aryl or hetaryl substituents (Figure 46) [84].

The compounds were tested for their antibacterial activity against Gram-positive and Gram-negative strains. Compounds **248**–**253** showed superior activity (MICs = 0.004–0.0016 mM) against MRSA (*S. aureus* N315), *E. coli* ATCC 25922, and *E. faecalis*, compared to norfloxacin (MICs = 0.025–0.050 mM) [84].

SAR studies of these series of compounds underline the selectivity confined by the two major substituents, phenyl and indole, towards MRSA, *E. coli*, and *E. faecalis* (Figure 46) [84]. The substituents on these rings influence the antibacterial activity. In the phenyl series, the 4-nitro (**249**) and 2,4-dichloro (**248**) groups influenced activity towards MRSA, while 4-methyl (**251**) influenced activity against *E. coli*. The 4-methoxy (**250**) substitution was the most advantageous against both strains [84].

In the indole series, small alkyl chains (**252** and **253**) selectively influenced activity against *E. faecalis*. As reported by the authors, the bis-*N*-(2-hydroxyethyl) groups are important for the development of some antiparasitic and antifungal drugs (Figure 46) [84].

These compounds can express antibacterial activity by targeting DNA gyrase, based on molecular docking studies. The proposed target was further confirmed through a biological assay, which showed that compound **250** could insert itself in MRSA DNA and inhibit its replication [84].

In conclusion, the clubbing of thiazole with coumarin resulted in very potent antibacterial and antifungal agents against a broad spectrum of pathogen strains, even including MRSA and mycobacteria.

This broad spectrum was also demonstrated in other hybrid compounds bearing a coumarin heterocycle, previously mentioned in this paper [28,31,32,33,57].

The coumarin heterocycle should be taken into consideration when designing novel antibacterials against problematic strains, like MRSA.

#### 2.5.2. Thiazolyl–Flavone Hybrid Compounds

Flavones are part of the large family of flavonoids, a widely distributed class of natural polyphenolic compounds. These impressive compounds exhibit a large variety of biological activities, including antimicrobial, antioxidant, and anticancer activities, thus constituting an attractive scaffold for drug design [85].

Zhao et al. reported the design and synthesis of a series of antistaphylococcal 2-(7-aminoethoxyflavonyl)-thiazoles, substituted on the fifth position of the thiazole ring with various aryl substituents (Figure 47) [86].

The compounds were tested for their antistaphylococcal activity against 40 clinical isolates (DPHS001 to DPHS040) of *S. aureus*, one MRSA strain (ATCC 43300), and one MSSA strain (ATCC 29213). Out of the 40 clinical isolates, 35 were mecA-positive (DPHS001 to DPSH035), (mecA = the gene complex responsible for altered penicillin-binding protein (PBP2a) synthesis and methicillin resistance), and 5 were mecA-negative (DPHS036 to DPHS040) [86].

Only three compounds (**254**–**256**) were active against the clinical isolates tested. Antibacterial activity was higher against MSSA and mecA-negative strains (MICs = 31.2–125 µg/mL) than against MRSA and mecA-positive strains (MICs = 31.2–500 µg/mL. The most active was compound **254**, with good activity against all the tested strains (MICs = 31.2–125 µg/mL) [86].

SAR studies of these compounds showed that, for the nitro derivatives, 4-nitro substitution (**256**) increased antistaphyloccocal activity, while 2- and 3-nitro substitutions inactivated the compounds (Figure 47) [86]. For the chloro derivatives, 2-chloro substitution (**254**) induced activity against all the tested strains, while 3-chloro (**255**) only induced activity against MSSA and mecA-negative strains. 4-Chloro and other substituents, such as fluoro, hydroxy, and methyl, inactivated the compounds [86].

These compounds expressed their antistaphylococcal activity by targeting DNA gyrase. Their inhibitory capacity was evaluated in comparison to novobiocin (IC_50_ = 0.8–3.23 µg/mL) [86]. Six compounds presented a promising inhibitory capacity: IC_50_ = 23.45–38 µg/mL against DPHS001 DNA gyrase and IC_50_ = 12.21–31.34 µg/mL against DPHS0036 DNA gyrase. Molecular docking studies showed that the 4-nitro derivative **256** was able to block the ATP binding site of DNA gyrase [86].

#### 2.5.3. Thiazolyl–Quinoline and Thiazolyl–Quinolone Hybrid Compounds

Quinoline and quinolinone are heterocycles found in some antimalarial drugs, such as quinine, and antibacterial compounds, as part of the fluoroquinolones class. Thus, they should be taken into consideration when designing novel antimicrobial compounds [87]. Herein, we report the structure–activity relationship of antimicrobial thiazoles clubbed with quinolines and quinoline-2(1*H*)-ones.

Ammar et al. reported the synthesis of some 2-(quinolin-3-yl-methylenehydrazinyl)-thiazoles, substituted in the fourth and fifth positions of the thiazole ring (Figure 48) [88].

The compounds were tested for their antibacterial activity against sensitive and multidrug-resistant bacterial strains and against some fungal strains [88]. Compounds **257**–**259** showed superior activity (MICs = 0.97–27.77 µg/mL) against *S. aureus*, *E. faecalis*, *E. coli*, *P. aeruginosa*, and *S. typhi* sensitive strains, compared to tetracycline (MICs = 15.62–62.5 µg/mL) [88]. Regarding the multidrug resistant strains, these compounds showed similar activity against *S. aureus* (ATCC 33591), *E. coli* (BAA-196), and *P. aeruginosa* (BAA-2111), compared to vancomycin (MICs = 0.78–1.95 µg/mL) and ciprofloxacin (MICs = 1.38–3.9 µg/mL) [88].

Concerning antifungal activity, compounds **257**–**259** showed similar activity (MICs = 7.81–62.5 µg/mL) against *C. albicans* and *F. oxysporum*, compared to amphotericin B (MICs = 15.62–31.25 µg/mL) [88].

SAR studies of these compounds underline the importance of a substituent in para position of the phenyl ring from the fourth position of the thiazole ring, as the unsubstituted thiazole compound had low antibacterial and antifungal activities (Figure 48) [88]. The nature of the substituent was as important as well, since the hydroxy derivative (**258**) had better activity compared to the chloro derivative (**257**). The importance of electron-donating groups can be seen in compound **259**, where the phenyl ring was replaced with methyl [88].

These compounds express their antibacterial activity by targeting DHFR. Compound **258** showed a better inhibitory capacity (IC_50_ = 10.02 ± 1.05 µM) than trimethoprim (IC_50_ = 18.25 ± 0.65 µM) [88].

Litim et al. designed some 2-(quinolin-3-yl)- and 2-(quinolinon-3-yl)-thiazoles, in which the heterocycles are linked by an α-aminophosphonate group (Figure 49). The compounds obtained are substituted on the hexatomic heterocycles with methyl or methoxy groups and on the fifth position of the thiazole ring with a phenyl or coumarinyl ring. Additionally, the quinoline derivatives are substituted on the third position with hydroxy or chloro substituents [89].

The compounds were tested for their antibacterial activity against sensitive and drug-resistant bacterial strains and for their antifungal activity against *C. albicans* [89]. Compounds **260**–**266** showed similar or lower activity (MICs = 0.25–128 µg/mL) against *S. aureus* ciprofloxacin-resistant (cipro R), ATCC 25923, and ATCC 6538, *E. faecalis* vancomycin-resistant (vanco R) and ATCC 29212, *E. coli* ATCC 25922, ESBL (extended spectrum beta-lactamase), cipro R, colistin-resistant (mcr-1), and ATCC 8739, *P. aeruginosa* VIM-2 and ATCC 9027, *A. baumanii* NDM-1 and OXA-23, *E. cloacae* fosfomycin-resistant (fos R), *K. pneumoniae* (carbapenem-resistant and carbapenem-sensitive), *S. marcescens*, *S. typhi* 14028, and *Citrobacter* sp., compared to imipenem, ciprofloxacin, and amikacin (MIC = 2 µg/mL) [89].

Concerning antifungal activity, compounds **260**–**266** showed similar or lower activity (MICs = 0.25–32 µg/mL) against *C. albicans* compared to fluconazole (MIC = 2 µg/mL) [89].

According to the SAR studies, quinolone compounds showed better activity (MICs = 0.25–256 µg/mL) than quinoline derivatives (MICs = 0.25–512 µg/mL) (Figure 49) [89]. The inhibitory effect was also related to the type of substituent on the α-aminophosphonate moiety, with 5-(3-coumarinyl)-thiazole being more active than the 5-phenylthiazole substituent [89].

The presence of both 2-hydroxyquinoline and 5-(coumarin-3-yl)-thiazole moieties in the same molecule (compounds **264**–**265**) favorized antibacterial activity against *E. faecalis* vanco R and *A. baumanni* NDM 1. The combination between quinolone and 5-phenylthiazole moieties (**263**) showed potential activity against *E. faecalis* ATCC 29212, *E. cloacae* fos R, and *K. pneumoniae* carbapenemase-producing bacteria, while the combination between quinolone and 5-(coumarin-3-yl)-thiazole moieties (**260**–**262**) favorized antibacterial activity against *E. faecalis* vanco R and *E. coli* ESBL. No potential target was reported by the authors [89].

#### 2.5.4. Thiazolyl–Quinazolone Hybrid Compounds

Quinazolones are found in compounds with a wide range of activities, including antibacterial, antifungal, anti-HIV, antimalarial, and antituberculosis activities [90]. Herein, we report the structure–activity relationship in antimicrobial thiazoles clubbed with quinazolones.

Desai et al. designed and synthesized antibacterial 2-(quinazolon-3-yl)-thiazole Mannich bases, directly linked to a substituted 5-thioxo-1,3,4-oxadiazole heterocycle (series **I**) as DNA gyrase inhibitors, as was predicted by molecular docking studies (Figure 50) [91].

In a similar manner, Wang et al. reported the identification of thiazole quinazolones, in which the two heterocycles are linked by an α,β-unsaturated carbonyl group (series **II**), as lactate dehydrogenase (LDH) inhibitors (Figure 50) [92].

The inactivation of LDH by compound **270** was supported by molecular docking studies. Important for the interaction with the target were the 7-chloro substituent, which formed hydrophobic bonds with Ile241 residue; the thiazole ring, which formed π-alkyl bonds with Ile25 and Val30 residues; and the two carbonyl groups and sulfur atom, which interacted through hydrogen bonds with Asn137 residue [92].

The compounds were tested for their antibacterial activity against Gram-positive and Gram-negative strains. Compounds **267**–**269** showed inferior activity (MICs = 50–150 µg/mL) against *S. aureus* MTCC-96, *E. coli* MTCC-443, and *P. aeruginosa* MTCC-1668, compared to gentamycin (MICs = 0.05–1 µg/mL) [91]. Compound **270** showed superior activity (MIC = 1 µg/mL) against *E. coli* and *P. aeruginosa* compared to norfloxacin (MIC = 4 µg/mL) [92].

Compound **270** produced inappreciable hemolysis, compared to the other tested compounds [92]. It showed a synergistic effect in combination with norfloxacin and displayed different mechanisms of action, including the blocking of biofilm formation, favorizing ROS generation in the bacterial cell and distortion of the bacterial metabolism by lactate dehydrogenase (LDH) inactivation and intercalating into the bacterial DNA, all of which could be responsible for low bacterial resistance [92].

SAR studies on the series **I** compounds showed that the presence of an electron-withdrawing group on the anilino moiety linked by a methylene bridge to the 1,3,4-oxadiazole ring favorized antibacterial activity (Figure 50) [91].

Based on the SAR studies on series **II**, 7-chloro substitution is important for the antibacterial effect, as well as an large electron-withdrawing group, like trifluoromethyl, linked to the benzylidene moiety (Figure 50) [92].

### 2.6. Thiazole Clubbed with Condensed Heterocycles

Condensed heterocycles are moieties that result from fusing two or more heterocycles into compact structures with enhanced properties. Based on the literature research, examples of condensed heterocycles clubbed with thiazole are quinuclidine, thiazolopyrimidines, pyrido-thiazolopyrimidines, and pyrazolo-thiazolopyrimidines [93,94]. Herein, we report the SAR studies of antimicrobial thiazole clubbed with condensed heterocycles.

#### 2.6.1. Thiazolyl–Quinuclidine Hybrid Compounds

Quinuclidine is found in the structures of natural compounds extracted from *Cinchona* species such as alkaloids, as well as in semisynthetic authorized antibacterials, namely, quinupristin [95].

Łączkowski et al. designed a series of 2-(quinuclidin-3-ylidene)-hydrazonothiazoles, with a large potential of biological activities, including antibacterial and antifungal ones. These compounds are substituted in the fourth position of the thiazole ring with various aryl substituents (Figure 51) [93].

The compounds were tested for their antibacterial activity against Gram-positive and Gram-negative strains and for their antifungal activity against *Candida* sp. strains [93]. Compounds **271**–**276** showed similar or inferior activity (MICs = 0.48–500 µg/mL) against *S. aureus* ATCC 6538, ATCC 25923, and ATCC 43300, *S. epidermidis* ATCC 12228, *M. luteus* ATCC 10240, *E. coli* ATCC 25922, *P. mirabilis* ATCC 12453, *K. pneumoniae* ATCC 13883, *S. thyphimurium* ATCC 14028, and *B. bronchiseptica* ATCC 4617, compared to ciprofloxacin (MICs = 0.004–0.98 µg/mL) [93].

Regarding antifungal activity, compounds **271**–**276** showed inferior activity (MICs = 1.95–125 µg/mL) against *C. albicans* ATCC 2091 and ATCC 10231, *C. parapsilosis* ATCC 22019, *C. glabrata* ATCC 90030, and *C. krusei* ATCC 14243, compared to nystatin (MICs = 0.24–0.48 µg/mL) [93].

According to SAR studies, substitutions with halogens (**271**, **272**, **274**, and **275**), large electron-withdrawing groups (**276**), and methyl (**273**) were the most advantageous for antibacterial and antifungal activities overall (Figure 51) [93]. The bromo (**274**) substitution induced the best antifungal activity, the iodo substitution (**275**) induced the best activity against Gram-positive bacterial strains, and the trifluoromethyl group (**276**) favorized antibacterial and antifungal activities overall. Substitution with larger groups, such as amides, sulfonamides, or large heterocycles like coumarin, decreased the activity, while additional substitutions on the thiazole ring canceled the effect [93].

These compounds express their antibacterial activity by targeting DNA gyrase and express their antifungal activity by targeting secretory aspartyl proteinase, important for fungal pathogenesis, based on molecular docking studies [93].

#### 2.6.2. Thiazole Clubbed with Polyheterocyclic Systems

An important strategy for the design of bioactive compounds is represented by the obtention of some polyheterocyclic systems, as they combine the biological activities of each of their components into compact structures.

Abdel-Latif et al. designed and synthesized thiazole-based polyheterocyclic systems, containing pyrimidine, pyridine, triazole, benzimidazole, and pyrazole heterocycles, as new antibacterial compounds (Figure 52) [94].

Starting from 3-cyano-thiazolo [3,2-*a*]pyrimidinone (**277**), they obtained some polyheterocyclic systems by annelation with pyrido[2′,3′:3,4]pyrazolo[1,5-*a*]pyrimidin-4-amine (**a**), benzo[4,5]imidazo[1,2-*a*]pyrimidin-4-amine (**b**), pyrazolo[1,5-*a*]pyrimidin-7-amine (**c**), [1,2,4]triazolo[3,4-*a*]pyrimidin-5-amine (**d**), 4,5-dihydro-1*H*-pyrazol-3-amine (**e**), and pyridines-2,6-diamine substituted in the ninth position (**f**, **278**–**279**). Other series contained isolated heterocycles as thiazol-2-yl-pyridinone derivatives (**g**) (Figure 52) [94].

The activity of the compounds was assayed against *S. aureus* and *E. coli*, using the agar diffusion method and a determination of the inhibition zone, compared to ampicillin (IZs = 24–27 mm, 100%) [94]. Compounds 2**61**–2**63** showed similar activity with the reference against *S. aureus* (IZs = 20–22 mm, 83.3–91.7%) and *E. coli* (IZs = 22–23 mm, 81.48–85.18%) [94].

Despite the heterogeneity of the synthesized molecules, some SAR studies can be underlined based on their antibacterial potential (Figure 52) [94]. The first difference can be noticed based on how the thiazole is linked to the rest of the system. Condensed thiazolo[3,2-*a*]pyrimidinones (**a**–**f**) displayed stronger activity than isolated thiazol-2-yl-pyridinones (**g**) [94].

Concerning the polycondensed heterocyclic systems, differences occur in the number of annulated rings. Activity decreased by supplementary annulation. The best results were obtained for the bicyclic compound **277** (IZ = 22 mm and 91.7% against *S. aureus*, while IZ = 23 mm and 85,18% against *E. coli*), with tetra- (**c**, **d**) and pentacyclic (**a**, **b**) compounds having very low or no activity (Figure 52) [94]. In the tricyclic compounds (**e**, **f**), pyrazolo-thiazolo[3,2-*a*]pyrimidinone (**e**) derivatives, containing azole rings, had lower activity compared to the pyrido-thiazolo[3,2-*a*]pyrimidinone (**f**) derivatives, containing azine rings (Figure 52) [94].

This activity was also influenced by the nature of the substituents present in the pyrido-thiazolo[3,2-*a*]pyrimidinone derivatives. Compound **278**, containing a 9-cyano substituent, displayed better activity (IZ = 21 mm and 87.5% against *S. aureus*, while IZ = 23 mm and 85.18% against *E. coli*) than compound **279**, containing a 9-ethoxycarbonyl substituent (IZ = 20 mm and 83.3% against *S. aureus*, while IZ = 22 mm and 81.48% against *E. coli*) (Figure 52). No potential target was reported by the authors [94].

## 3. Limitations

The limitations of this paper include the fact that the SAR studies were based on antimicrobial compounds—for which the activity was quantified using only disk diffusion—and the lack of SAR studies on additional activities, such as virulence factors, antibiofilm activity, or cytotoxicity. The disk diffusion method is currently considered an inaccurate assay to quantify antimicrobial activity, due to the different diffusion rates on agar plates of each given compound.

No extensive SAR studies on additional assays, such as the effects of compounds on different virulence factors, antibiofilm activity, or cytotoxicity were taken into account because these assays were performed. As these were only performed on a limited number of compounds, therefore, no structure–activity relationship could be depicted.

## 4. Conclusions

Thiazole is an important heterocycle for drug design, not only for antibacterials and antifungals but also many other pharmacological classes. A rational design approach to novel drug development allows for faster progress and effectiveness. Therefore, a rational approach for drug discovery requires optimization based on structure–activity relationships.

Based on the observations drawn in this review, antimicrobial activity is coherent with the chemical structure of the compounds reviewed. Thus, this work aimed to collect and visualize SAR studies of some of the reported antimicrobial scaffolds of hybrid thiazolic compounds as a means to observe the structural heterogeneity that can induce antimicrobial activity.

Concerning antimicrobial activity, we can conclude that thiazole clubbed with various heterocycles had different results in terms of its activity spectrum and potency, depending on the other heterocycles from the structure. Pyrazoline-clubbed antimicrobials expressed activity against Gram-positive bacteria and *K. pneumoniae*, while its activity was extended against fungal strains when using pyrazole, similar to imidazole. Hybrid thiazoles clubbed with thiazolidinone contained promising antituberculosis compounds, while clubbing with triazole covered all three types of activity: antibacterial, antifungal, and antituberculosis. Lower potency was observed in hybrid compounds with pyridine, but this was not transposable to the benzofused heterocycles, quinoline and quinolone. Similar to triazole, coumarin is another veritable scaffold for designing novel antimicrobials. Also, high importance should be allocated to the substituents used in each scaffold.

The observations made in this work could provide valuable scientific material in the design of novel antimicrobials in future research, which is also supported by the vast number of previously published papers referring to this topic of interest.

## Figures and Tables

**Figure 1 pharmaceutics-16-00089-f001:**
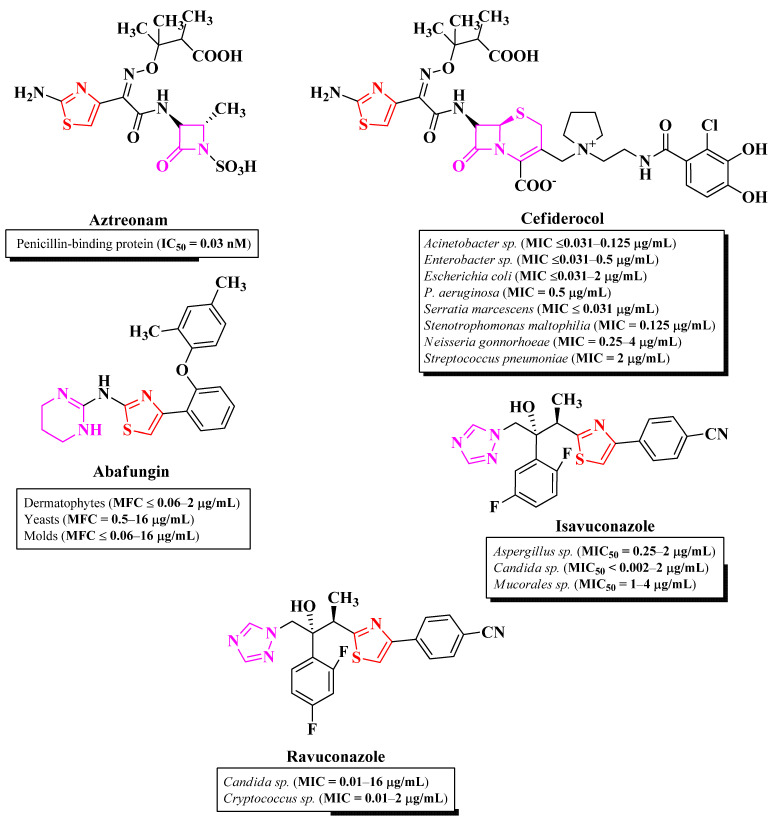
Some clinically authorized antimicrobial drugs containing the thiazole heterocycle clubbed with other heterocycles, with their potencies on given pathogens [7,8,9,10,11]. Legend: IC = inhibitory concentration; MIC = minimal inhibitory concentration; MFC = minimal fungicidal concentration.

**Figure 2 pharmaceutics-16-00089-f002:**
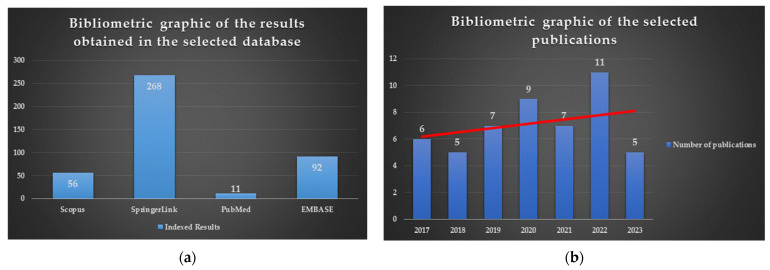
(**a**) Bibliometric graphic of the indexed results obtained from Scopus, SpringerLink, PubMed, and EMBASE databases; (**b**) Bibliometric graphic of the selected publications for this review, based on the publication year. The red line represents the increasing interest in the topic of thiazoles in antimicrobial hybrid compounds.

**Figure 3 pharmaceutics-16-00089-f003:**
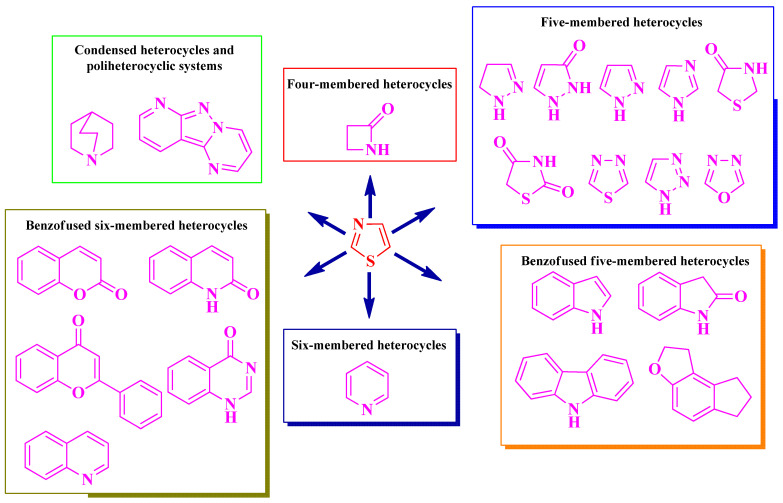
Summary of the antimicrobial thiazole-based hybrid compounds from the selected papers.

**Figure 4 pharmaceutics-16-00089-f004:**
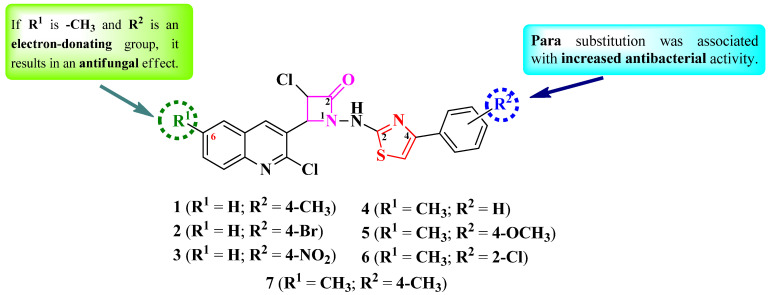
SAR studies of antimicrobial 4-(quinolin-3-yl)-1-(thiazol-2-yl)-amino-azetidin-2-one derivatives, reported by Desai et al. [13].

**Figure 5 pharmaceutics-16-00089-f005:**
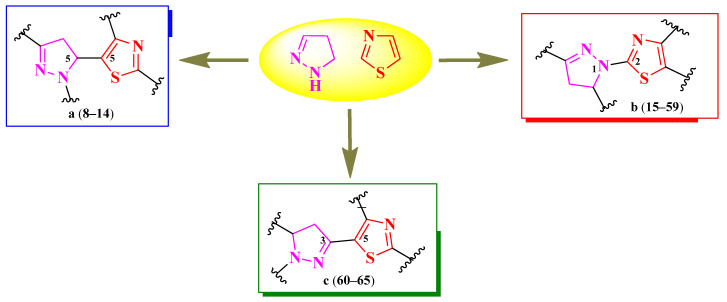
The general structures for the series discussed, containing thiazole and 2-pyrazoline rings: 5-(2-pyrazolin-5-yl)-thiazoles (**a**), 2-(2-pyrazolin-1-yl)-thiazoles (**b**), and 5-(2-pyrazolin-3-yl)-thiazoles (**c**) [15,16,17,18,19,20,21,22,23,24].

**Figure 6 pharmaceutics-16-00089-f006:**
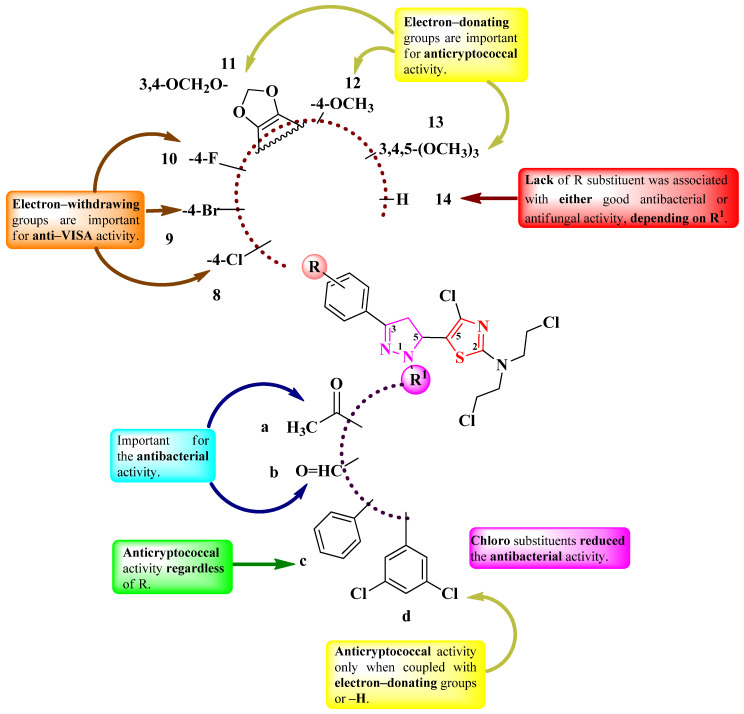
SAR studies of antimicrobial 2-(*N-*mustard)-5-(2-pyrazolin-5-yl)-thiazole derivatives, reported by Cuartas et al. [15].

**Figure 7 pharmaceutics-16-00089-f007:**
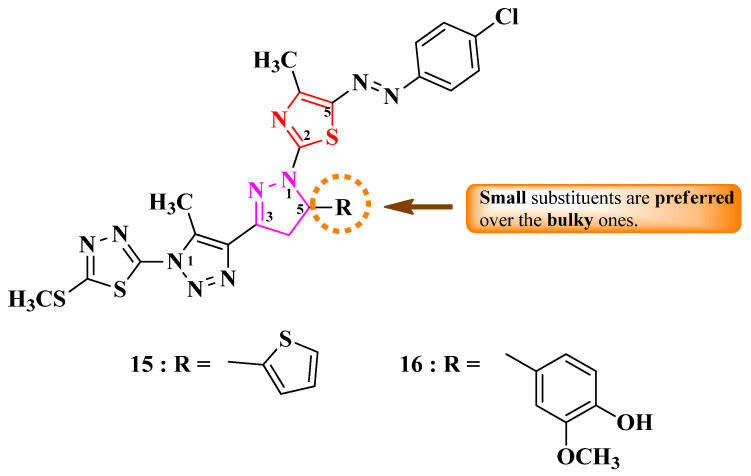
SAR study of antimicrobial 2-(4-(1-thiazol-2-yl)-2-pyrazolin-3-yl)-1,2,3-triazol-1-yl)-1,3,4-thiadiazole derivatives, reported by Rashdan and Abdelmonsef [16].

**Figure 8 pharmaceutics-16-00089-f008:**
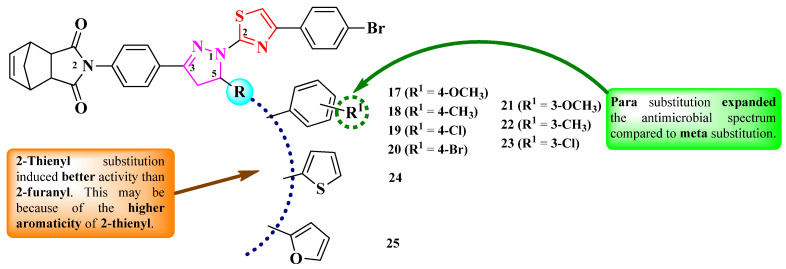
SAR studies of antimicrobial 2-(4-(1-(thiazol-2-yl)-2-pyrazolin-3-yl)-phenyl)-methano-isoindol-1,3-dione derivatives, reported by Budak et al. [17].

**Figure 9 pharmaceutics-16-00089-f009:**
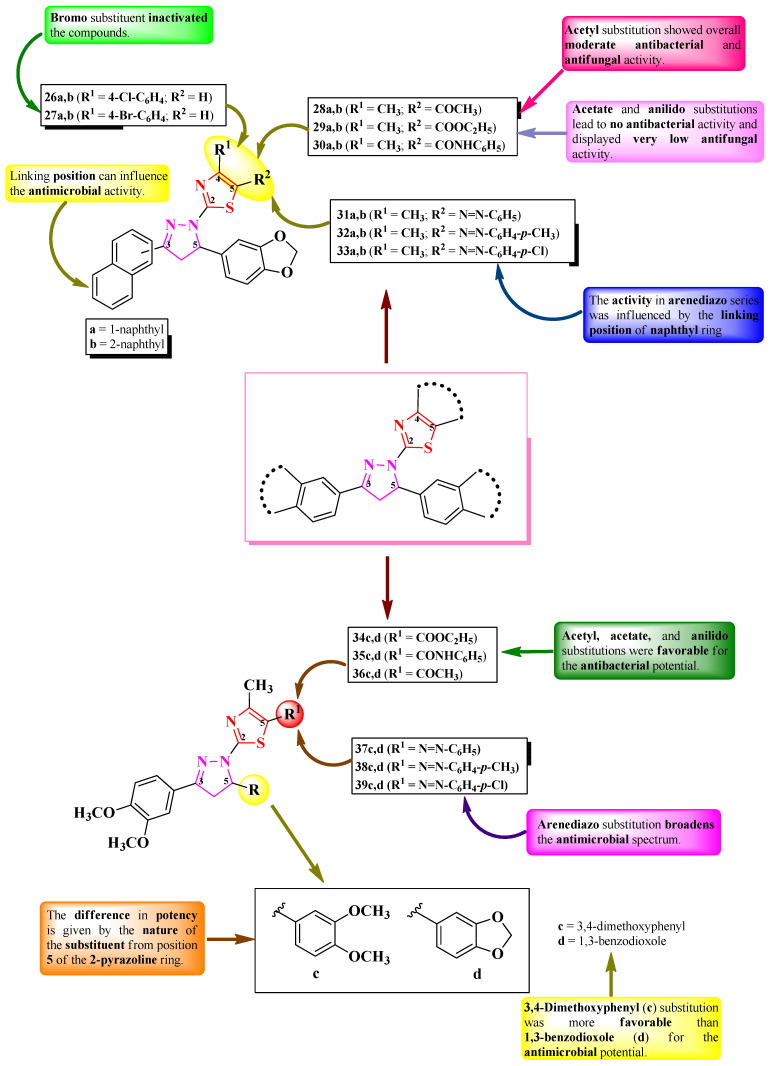
SAR studies of the series of antimicrobial 2-(3-aryl-5-hetaryl-2-pyrazolin-1-yl)-thiazole derivatives, synthesized by Mansour et al. [18] and Masoud et al. [19].

**Figure 10 pharmaceutics-16-00089-f010:**
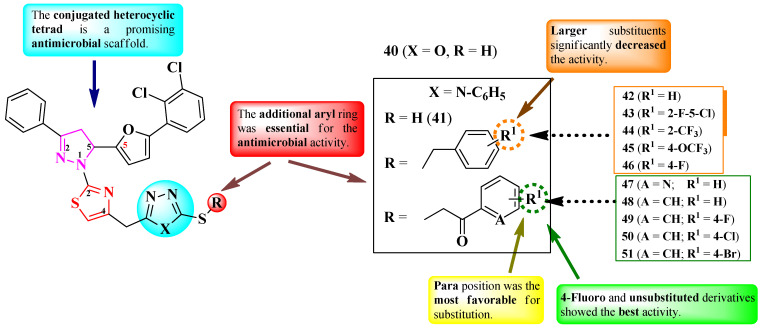
SAR studies of antimicrobial 2-(5-(furan-2-yl)-2-pyrazolin-1-yl)-4-methylhetarylthio-thiazole derivatives, reported by Bhandare et al. [20].

**Figure 11 pharmaceutics-16-00089-f011:**
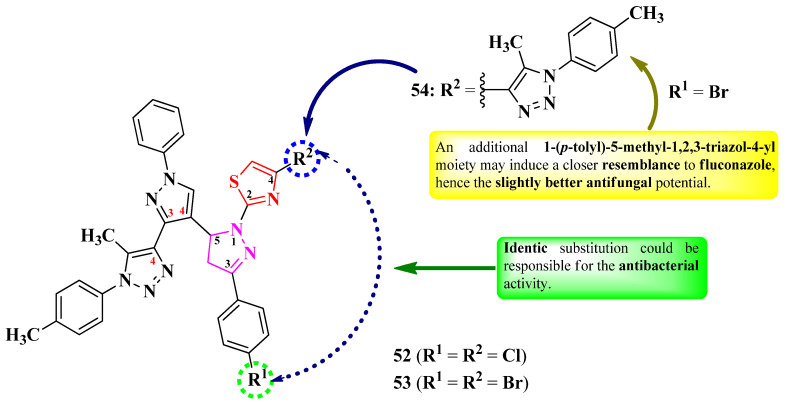
Structure–activity relationships in antimicrobial 2-(5-(3-(1,2,3-triazol-4-yl)-pyrazol-4-yl)-2-pyrazolin-1-yl)-thiazole derivatives, reported by Abdel-Wahab et al. [21].

**Figure 12 pharmaceutics-16-00089-f012:**
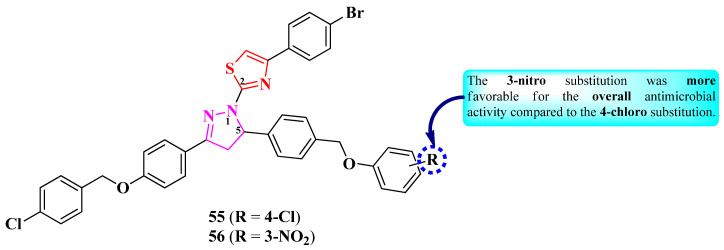
Antimicrobial 2-(2-pyrazolin-1-yl)-thiazole compounds, reported by Salih et al. [22].

**Figure 13 pharmaceutics-16-00089-f013:**
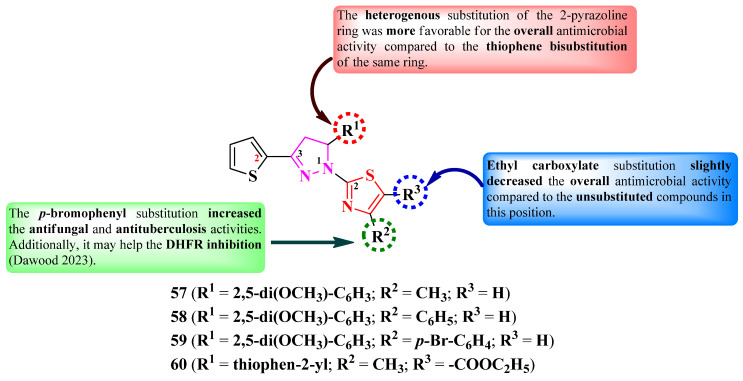
SAR studies of antimicrobial 2-(3-(thiophen-2-yl)-2-pyrazolin-1-yl)-thiazole derivatives, reported by Dawood et al. [23].

**Figure 14 pharmaceutics-16-00089-f014:**
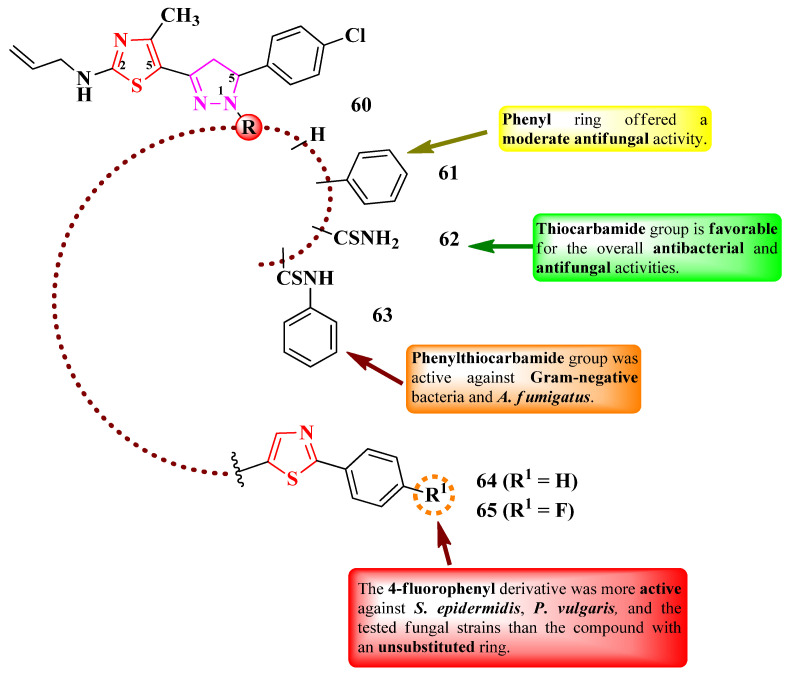
SAR studies of antimicrobial 2-(*N*-allyl)-5-(2-pyrazolin-3-yl)-thiazole derivatives, reported by Bondock and Fouda [24].

**Figure 15 pharmaceutics-16-00089-f015:**
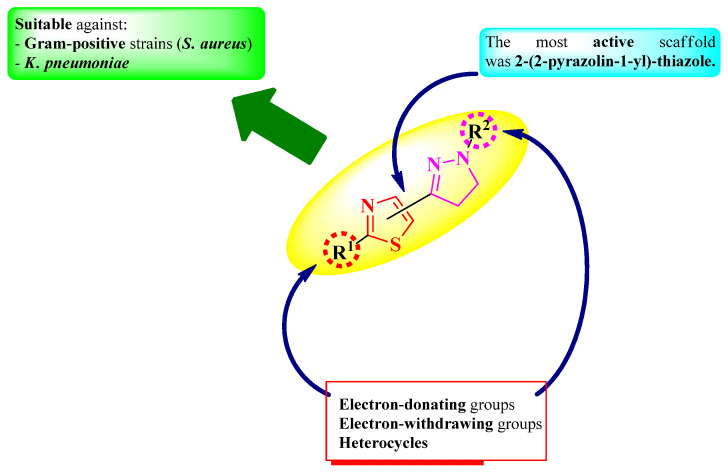
Endpoints of designing novel antimicrobial thiazole clubbed with 2-pyrazoline hybrid compounds.

**Figure 16 pharmaceutics-16-00089-f016:**
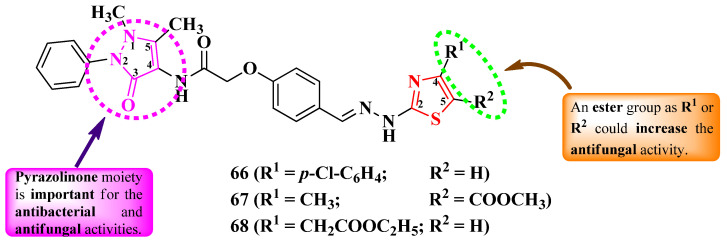
Antimicrobial *N*-(4-pyrazolin-3-one)-2-thiazolyl-hydrazonomethyl-phenoxyacetamides, reported by Abu-Melha [26].

**Figure 17 pharmaceutics-16-00089-f017:**
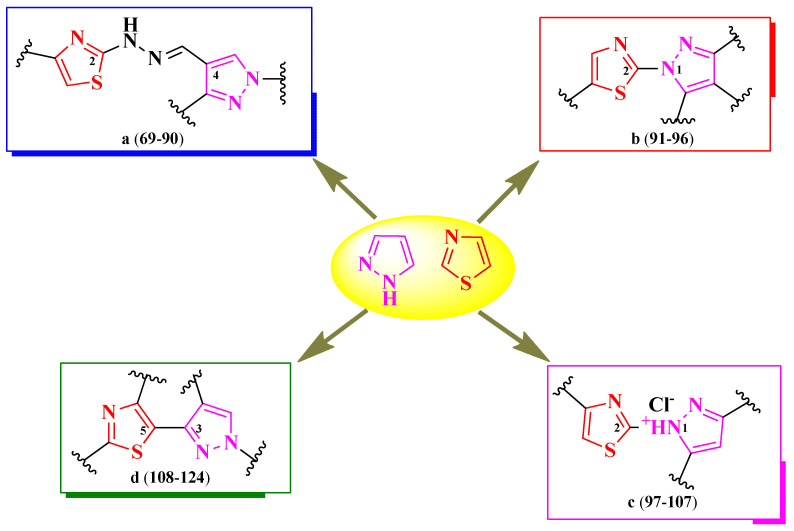
The general structures for the series discussed, containing thiazole and pyrazole rings: 2-(pyrazol-4-yl)-methylylidenehydrazinyl-thiazoles (**a**), 2-(pyrazol-1-yl)-thiazoles (**b**), 2-(pyrazol-1-ium-1-yl)-thiazoles (**c**), and 5-(pyrazol-3-yl)-thiazoles (**d**) [28,29,30,31,32,33,34].

**Figure 18 pharmaceutics-16-00089-f018:**
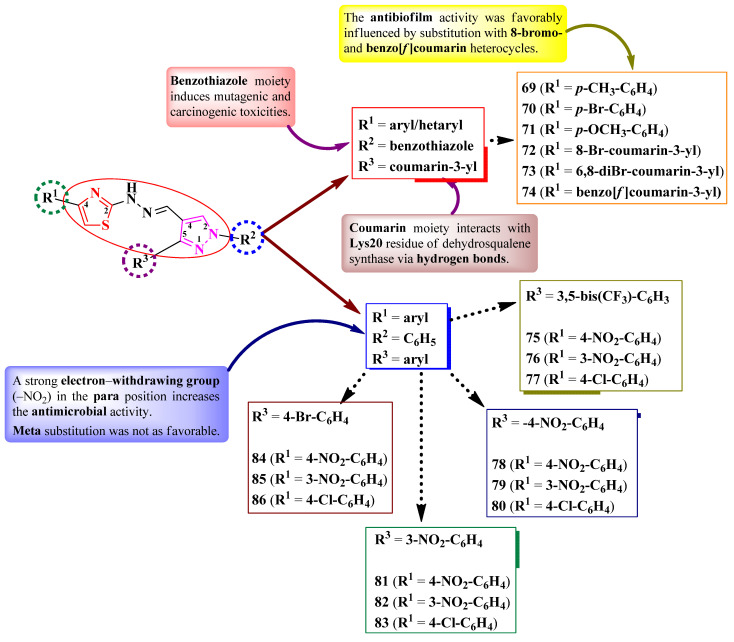
SAR studies of antimicrobial 2-(pyrazol-4-yl)-methylylidenehydrazinyl-thiazole derivatives, reported by Gondru et al. [28] and Patil et al. [29].

**Figure 19 pharmaceutics-16-00089-f019:**
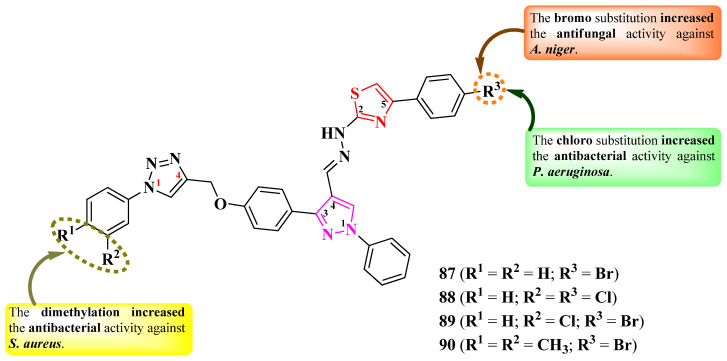
SAR studies of antimicrobial 2-(pyrazol-4-yl)-methylylidenehydrazinyl-thiazole derivatives, reported by Matta et al. [30].

**Figure 20 pharmaceutics-16-00089-f020:**
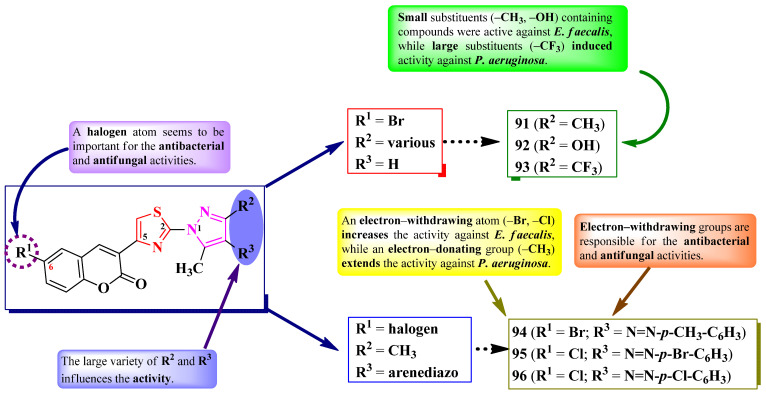
SAR studies of antimicrobial 5-(coumarin-3-yl)-2-(pyrazol-1-yl)-thiazole derivatives, reported by Abdel-Aziem et al. [31] and Kumar et al. [32].

**Figure 21 pharmaceutics-16-00089-f021:**
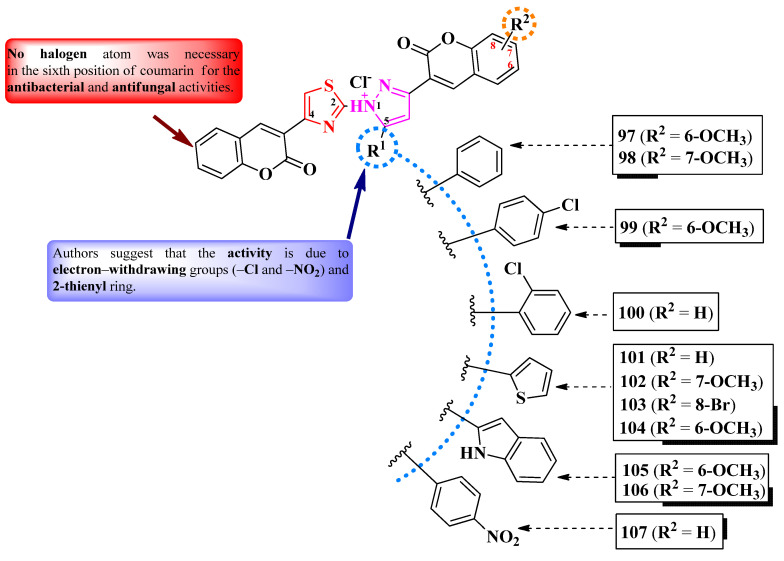
Antimicrobial 2-pyrazolium-thiazol-4-yl salts, reported by Mahmoodi and Ghodsi [33].

**Figure 22 pharmaceutics-16-00089-f022:**
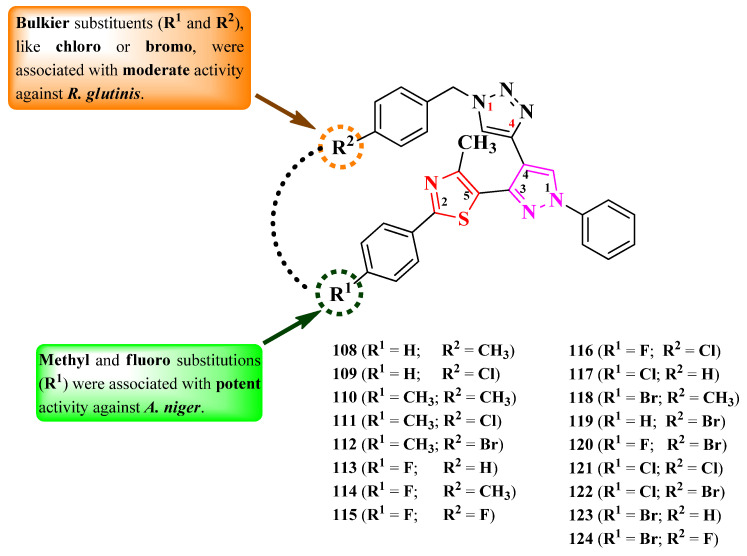
SAR studies of antimicrobial 2-phenyl-5-(4-hetaryl-pyrazol-3-yl)-thiazoles, reported by Nalawade et al. [34].

**Figure 23 pharmaceutics-16-00089-f023:**
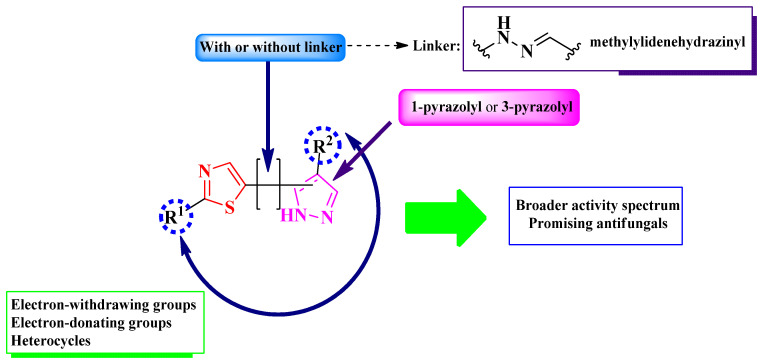
Endpoints of designing novel antimicrobial thiazole clubbed with pyrazole hybrid compounds.

**Figure 24 pharmaceutics-16-00089-f024:**
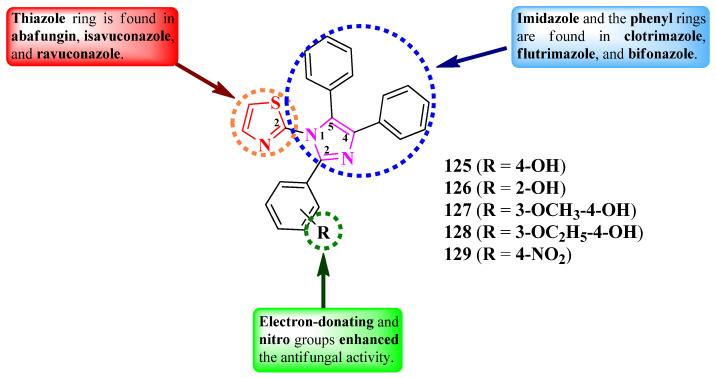
Design of antifungal 2-(2,4,5-triphenyl-imidazol-1-yl)-thiazoles, reported by Nikalje et al. [38].

**Figure 25 pharmaceutics-16-00089-f025:**
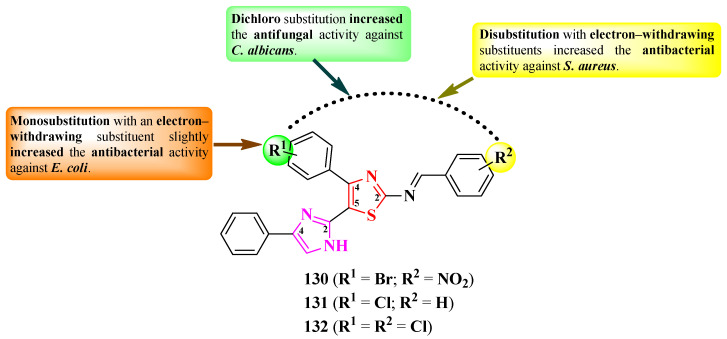
SAR studies of antimicrobial 2-(iminobenzylidenel)-5-(imidazole-2-yl)-thiazole derivatives, reported by Dekate et al. [39].

**Figure 26 pharmaceutics-16-00089-f026:**
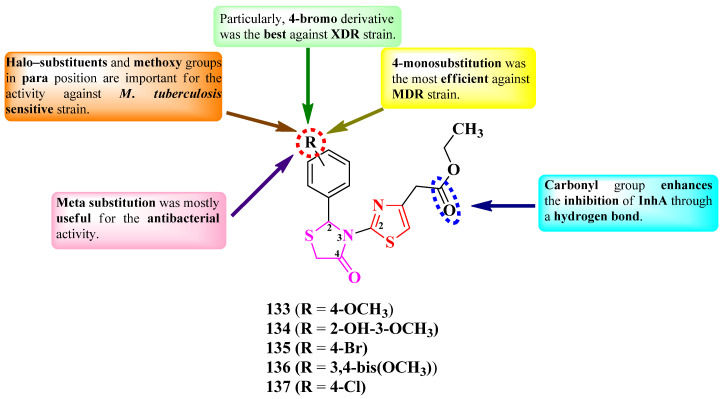
SAR studies of antibacterial 2-(thiazol-2-yl)-*N*-thiazolidin-4-one derivatives, reported by Othman et al. [41].

**Figure 27 pharmaceutics-16-00089-f027:**
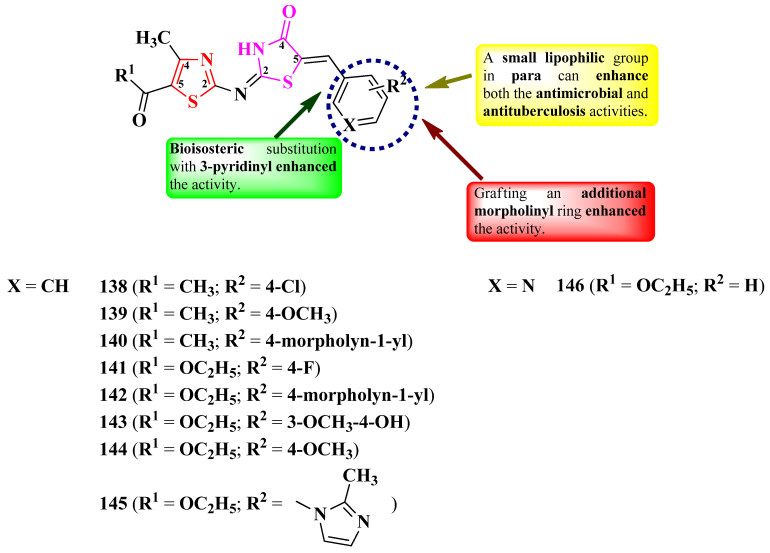
SAR studies of antimicrobial 2-(thiazol-2-yl)-imino-thiazolidin-4-one derivatives, reported by Abo-Ashur et al. [42].

**Figure 28 pharmaceutics-16-00089-f028:**
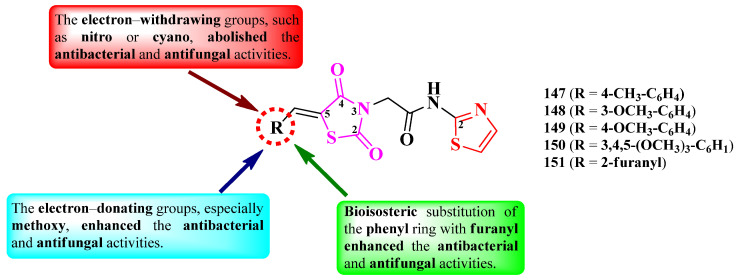
SAR studies of antimicrobial 2-(thiazolidin-2,4-dion-3-yl)-*N*-(thiazol-2-yl)acetamides, reported by Alegaon et al. [44].

**Figure 29 pharmaceutics-16-00089-f029:**
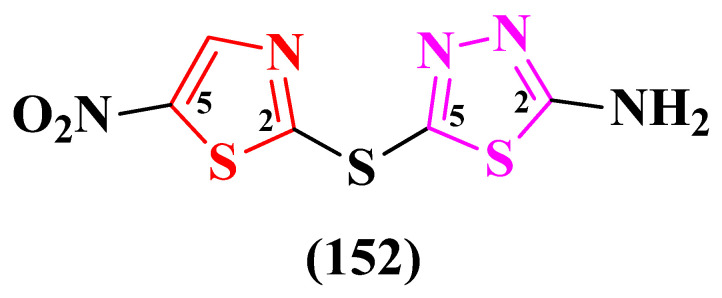
Structure of halicin.

**Figure 30 pharmaceutics-16-00089-f030:**
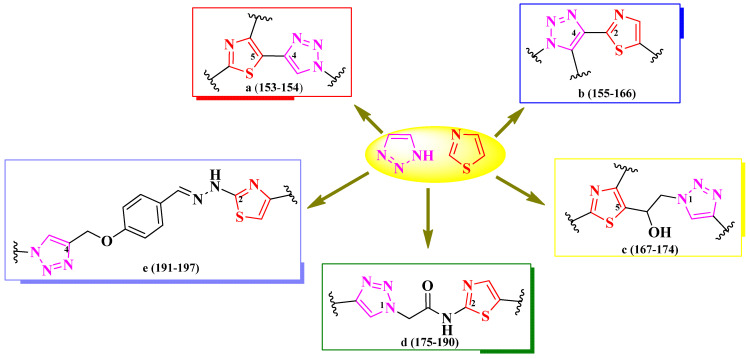
The general structures for the series discussed, containing thiazole and 1,2,3-triazole rings: 5-(1,2,3-triazol-4-yl)-thiazoles (**a**), 2-(1,2,3-triazol-4-yl)-thiazoles (**b**), 1-(thiazol-5-yl)-2-(1,2,3-triazol-1-yl)-ethanol derivatives (**c**), 2-(1,2,3-triazol-1-yl)-*N*-(thiazol-2-yl)-acetamides (**d**), and 2-(1,2,3-triazol-4-yl-methoxybenzylidenehydrazinyl)-thiazoles (**e**) [52,53,54,55,56,57].

**Figure 31 pharmaceutics-16-00089-f031:**
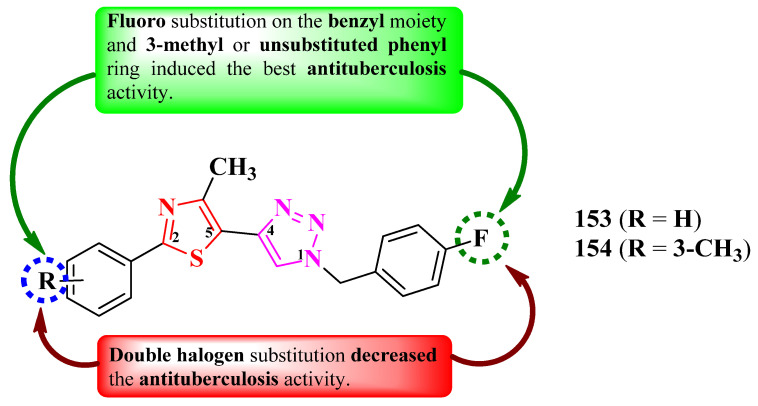
SAR studies of antituberculosis 5-(1,2,3-triazol-4-yl)-thiazoles, reported by Shinde et al. [52].

**Figure 32 pharmaceutics-16-00089-f032:**
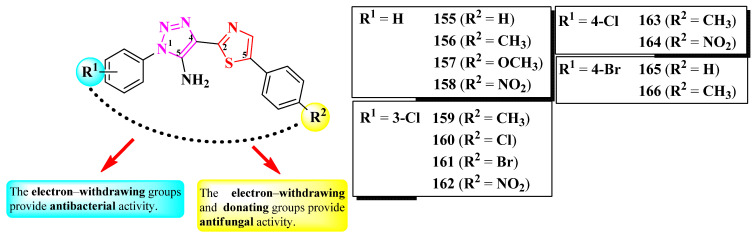
SAR studies of antimicrobial 2-(1,2,3-triazol-4-yl)-thiazole derivatives, reported by Mahale et al. [53].

**Figure 33 pharmaceutics-16-00089-f033:**
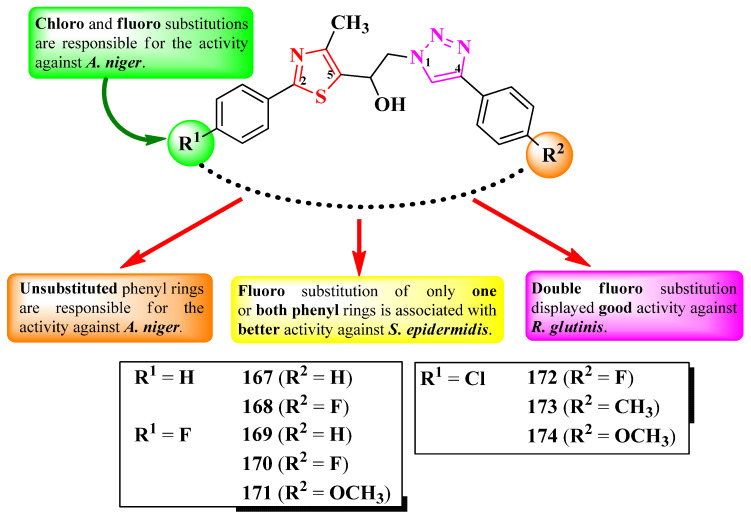
SAR studies of antimicrobial 1-(thiazol-5-yl)-2-(1,2,3-triazol-1-yl)-ethanol derivatives, reported by Jagadale et al. [54].

**Figure 34 pharmaceutics-16-00089-f034:**
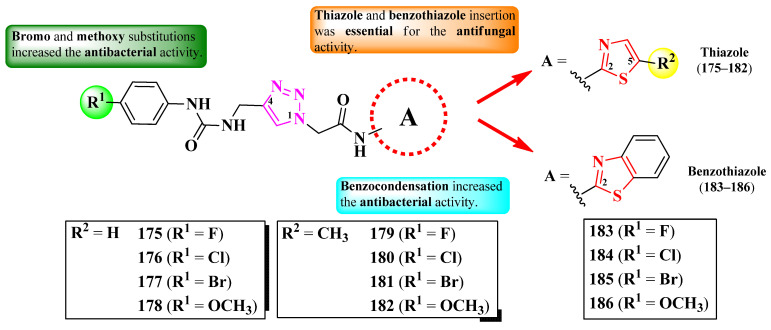
SAR studies of antimicrobial 2-(1,2,3-triazol-1-yl)-*N*-(thiazol-2-yl)-acetamides and 2-(1,2,3-triazol-1-yl)-*N*-(benzothiazol-2-yl)-acetamides, reported by Poonia et al. [55].

**Figure 35 pharmaceutics-16-00089-f035:**
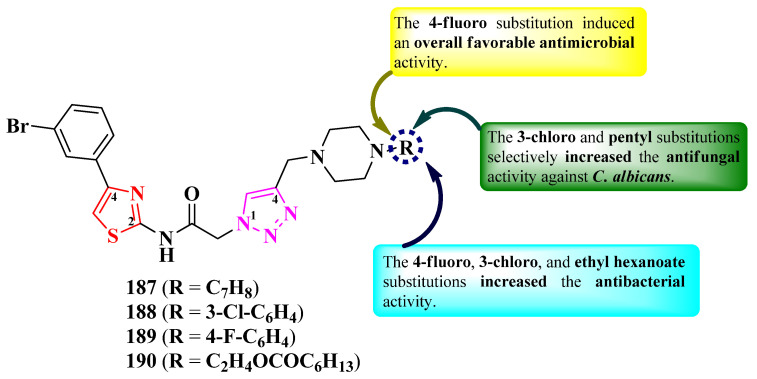
SAR studies of antimicrobial 2-(1,2,3-triazol-1-yl)-*N*-(thiazol-2-yl)-acetamides, reported by Veeranki et al. [56].

**Figure 36 pharmaceutics-16-00089-f036:**
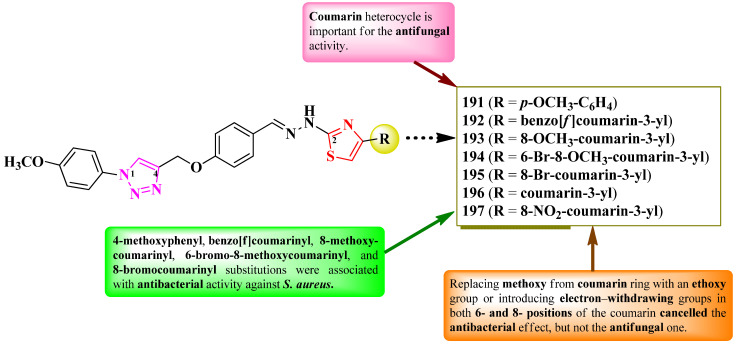
SAR studies of antimicrobial 2-(1,2,3-triazol-4-yl-methoxybenzylidenehydrazinyl)-thiazoles, reported by Gondru et al. [57].

**Figure 37 pharmaceutics-16-00089-f037:**
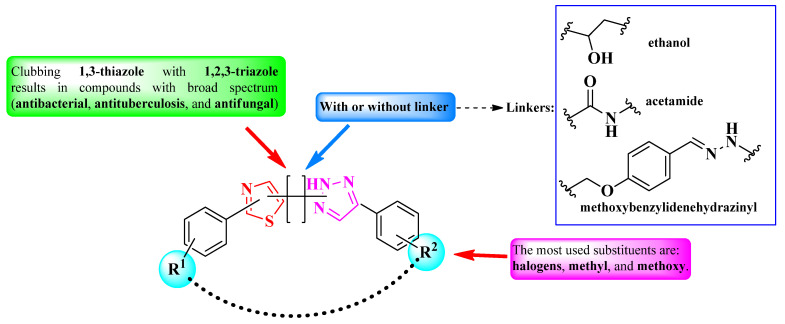
Endpoints of designing novel antimicrobial 1,3-thiazole clubbed with 1,2,3-triazole hybrid compounds.

**Figure 38 pharmaceutics-16-00089-f038:**
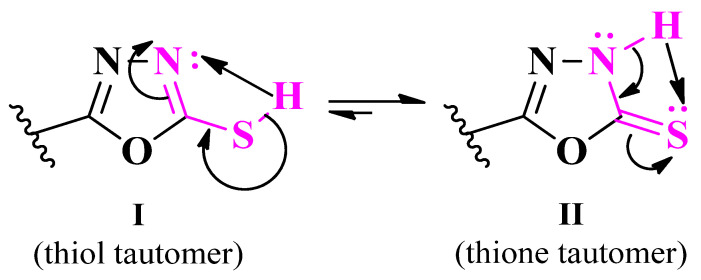
The thiol–thione tautomerism [64].

**Figure 39 pharmaceutics-16-00089-f039:**
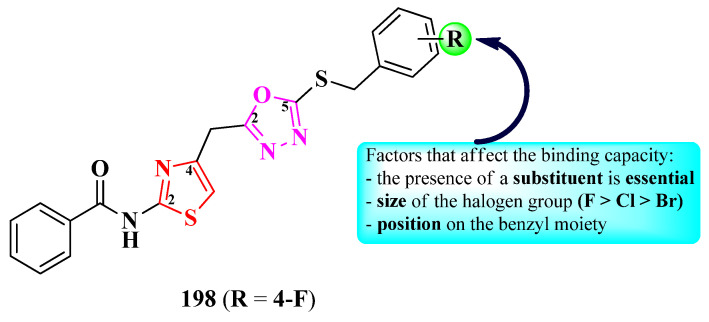
SAR studies of 1,3-thiazole clubbed in 2-(thiazol-4-yl)-methylene-5-thio-1,3,4-oxadiazoles urease inhibitors, reported by Athar Abasi et al. [62].

**Figure 40 pharmaceutics-16-00089-f040:**
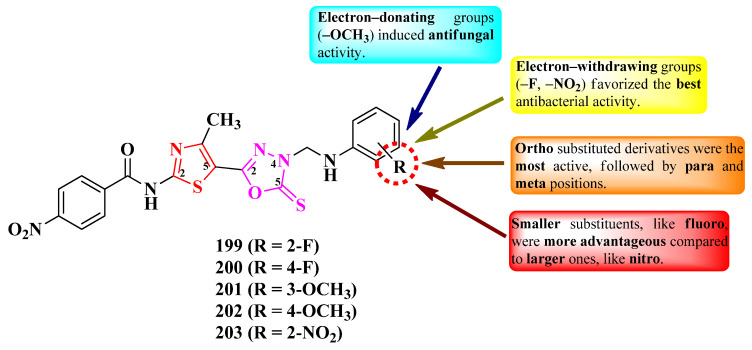
SAR studies of antimicrobial 2-(thiazol-5-yl)-5-mercapto-1,3,4-oxadiazoles, reported by Desai et al. [63].

**Figure 41 pharmaceutics-16-00089-f041:**
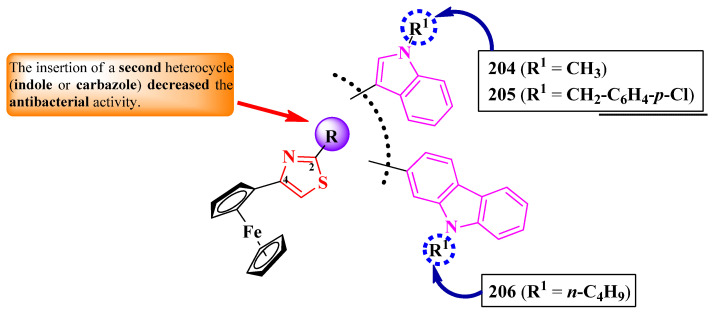
SAR studies of antibacterial 4-ferrocenylthiazoles clubbed with indole or carbazole, reported by Zhao et al. [70].

**Figure 42 pharmaceutics-16-00089-f042:**
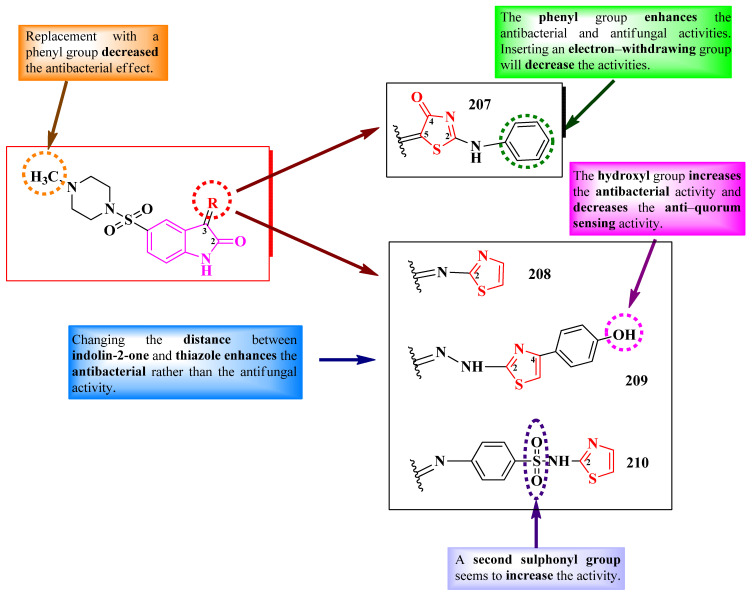
SAR studies of antimicrobial thiazole clubbed with indolin-2-one compounds, reported by Alzahrani et al. [72].

**Figure 43 pharmaceutics-16-00089-f043:**
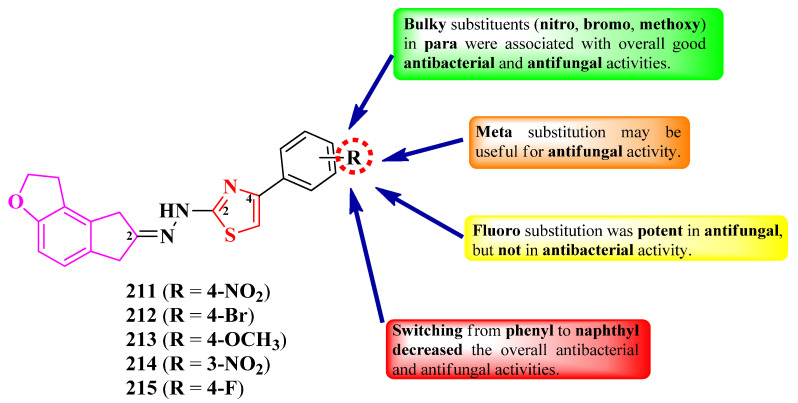
Structure–activity relationships in antimicrobial 2-(2-tetrahydroindenofuranylidene)-hydrazinylthiazoles, reported by Adole et al. [75].

**Figure 44 pharmaceutics-16-00089-f044:**
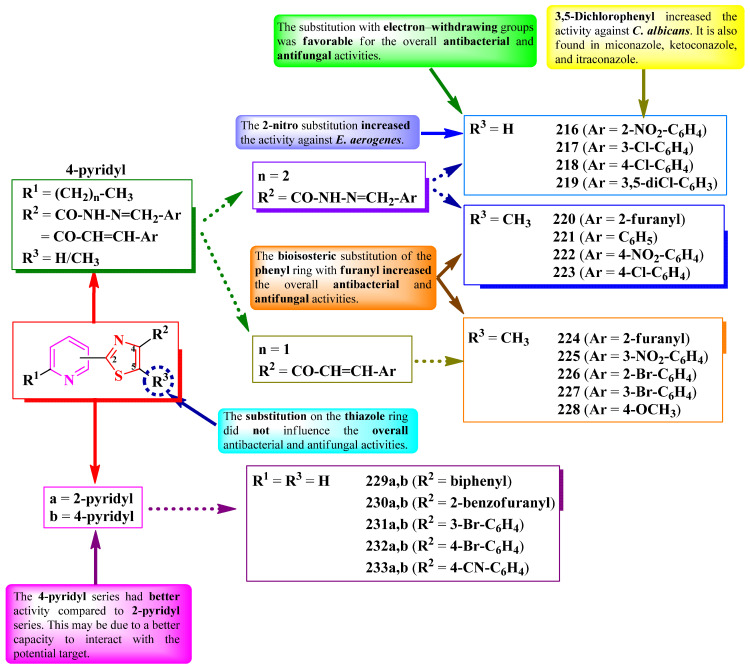
Structure–activity relationships in antimicrobial 2-pyridyl-thiazoles, reported by Muluk et al. [76,77], Patil et al. [78], and Eryılmaz et al. [79].

**Figure 45 pharmaceutics-16-00089-f045:**
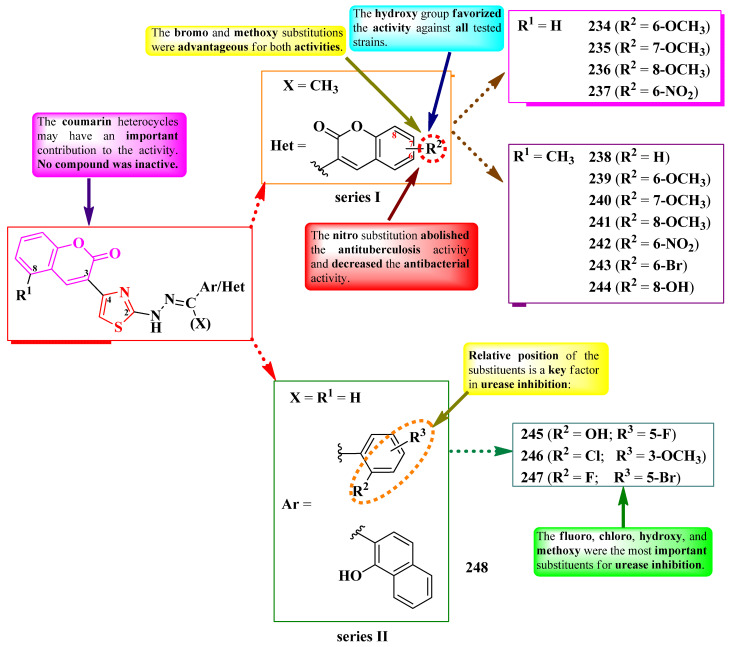
SAR studies of antimicrobial 4-(3-coumaryl)-2-hydrazinylthiazoles, reported by Yusufzai et al. [82] and Salar et al. [83].

**Figure 46 pharmaceutics-16-00089-f046:**
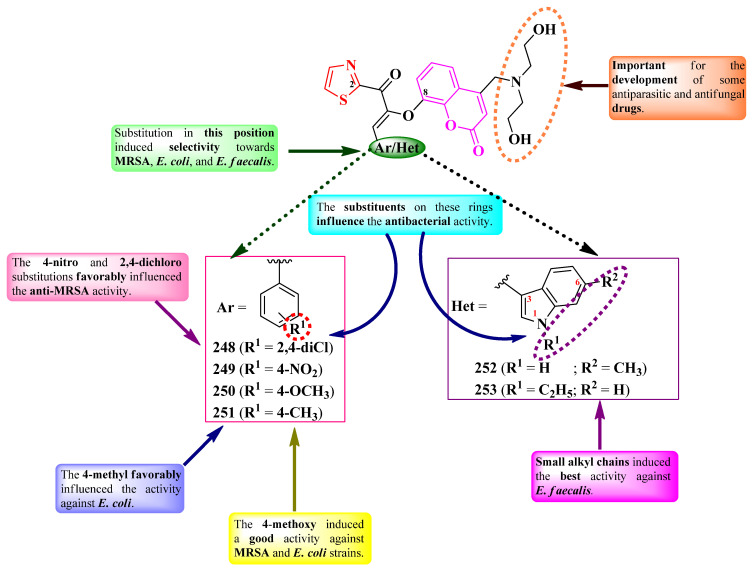
SAR studies of antibacterial 2-(8-(1-oxoprop-2-en-2-yl)-oxycoumarinyl)-thiazoles, reported by Hu et al. [84].

**Figure 47 pharmaceutics-16-00089-f047:**
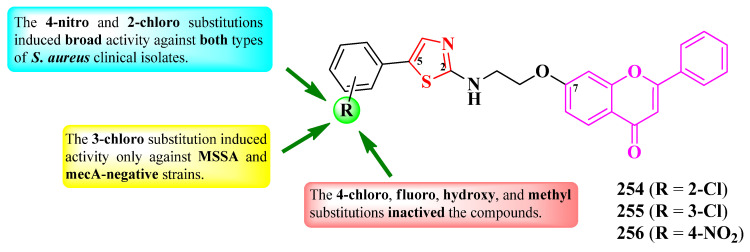
SAR studies of antistaphylococcal 2-(7-aminoethoxyflavonyl)-thiazoles, reported by Zhao et al. [86].

**Figure 48 pharmaceutics-16-00089-f048:**
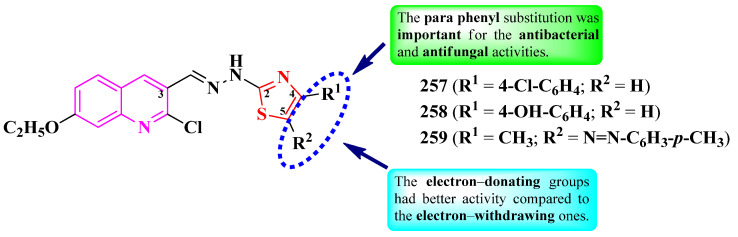
SAR studies of antimicrobial 2-(quinolin-3-yl-methylenehydrazinyl)-thiazoles, reported by Ammar et al. [88].

**Figure 49 pharmaceutics-16-00089-f049:**
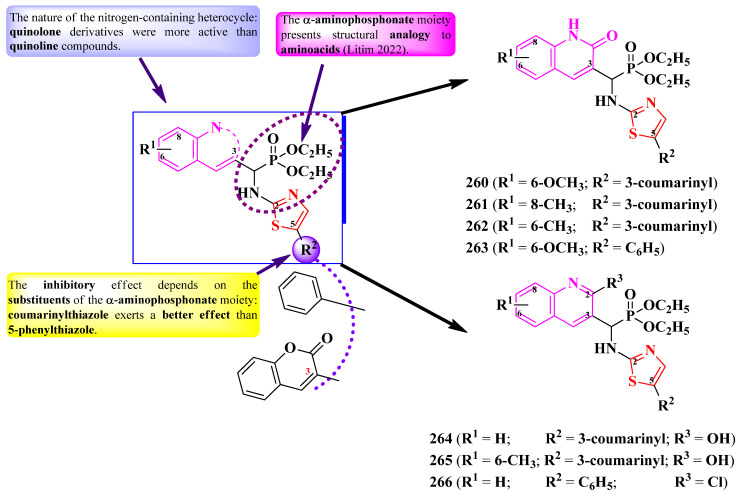
SAR studies of antimicrobial 2-(quinolin-3-yl)- and 2-(quinolinon-3-yl)-thiazoles, reported by Litim et al. [89].

**Figure 50 pharmaceutics-16-00089-f050:**
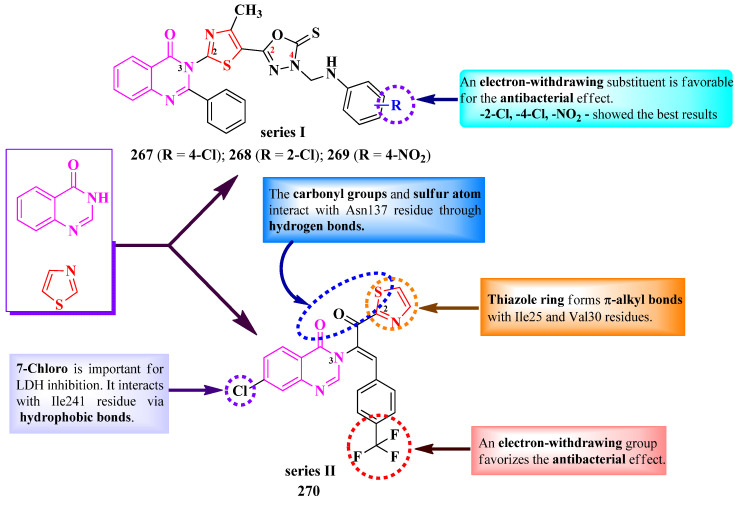
SAR studies of antibacterial quinazolonyl–thiazoles, reported by Desai et al. [91] and Wang et al. [92].

**Figure 51 pharmaceutics-16-00089-f051:**
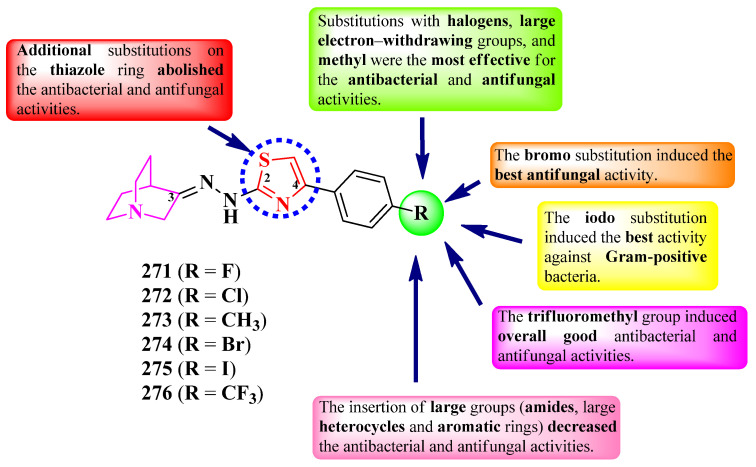
SAR studies of antimicrobial 2-(quinuclidin-3-ylidene)-hydrazonothiazoles, reported by Łączkowski et al. [93].

**Figure 52 pharmaceutics-16-00089-f052:**
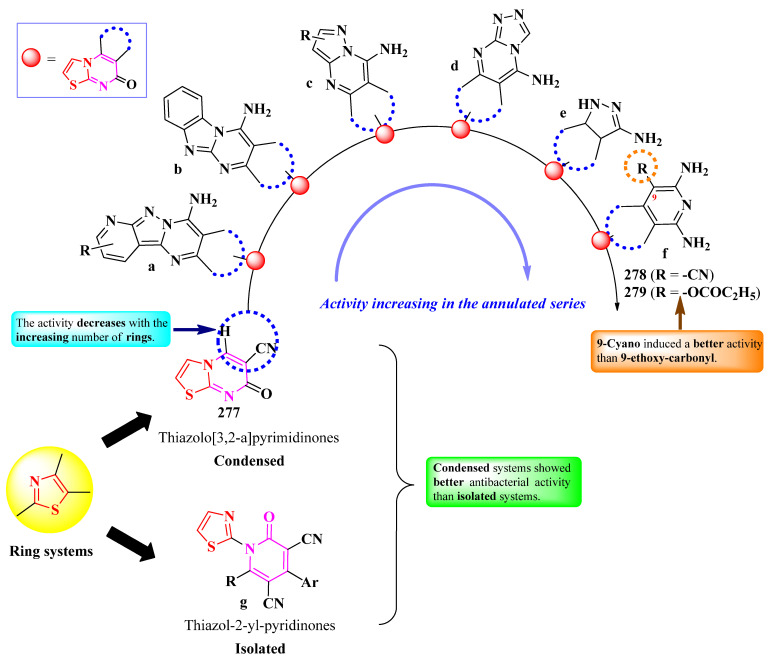
SAR studies of antibacterial 1,3-thiazole clubbed with polyheterocyclic systems, reported by Abdel-Latif et al. [94].

## Data Availability

The data presented in this study are available in this article.

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
