# Peer review of "An Overview of the Structure–Activity Relationship in Novel Antimicrobial Thiazoles Clubbed with Various Heterocycles (2017–2023)"

_pharmaceutics, 2024, doi:10.3390/pharmaceutics16010089_

Round 1

Reviewer 1 Report

Comments and Suggestions for Authors

Comments and Suggestions for Authors

The manuscript entitled " An Overview on the Structure-Activity Relationship in Novel Antimicrobial Thiazoles Clubbed with Various Heterocycles (2017-2022)" by Daniel Ungureanu and etal, study the structure-activity relationship in antimicrobial thiazoles clubbed with various heterocycles reported in the literature between 2017 and 2022.  

The manuscript is well-written and the finding is novel. Therefore it is appropriate that manuscript can be published in pharmaceutics after revision.

I have the following comments on the manuscript

-                In its current state, the level of English throughout your manuscript does not meet the journal's desired standard. Please check the manuscript and refine the language carefully.

-                There are some typographical errors, it should be considered

-                The rational of work should be expanded

-                Add some recent related references on bioactive thiazoles

Catalysts, 2023, 13, 1311.

Scientific Reports 2020, 10, 20863.

J. Heterocycl. Chem., 2018, 55(1), 258-264.

Comments on the Quality of English Language

Comments and Suggestions for Authors

The manuscript entitled " An Overview on the Structure-Activity Relationship in Novel Antimicrobial Thiazoles Clubbed with Various Heterocycles (2017-2022)" by Daniel Ungureanu and etal, study the structure-activity relationship in antimicrobial thiazoles clubbed with various heterocycles reported in the literature between 2017 and 2022.  

The manuscript is well-written and the finding is novel. Therefore it is appropriate that manuscript can be published in pharmaceutics after revision.

I have the following comments on the manuscript

-                In its current state, the level of English throughout your manuscript does not meet the journal's desired standard. Please check the manuscript and refine the language carefully.

-                There are some typographical errors, it should be considered

-                The rational of work should be expanded

-                Add some recent related references on bioactive thiazoles

Catalysts, 2023, 13, 1311.

Scientific Reports 2020, 10, 20863.

J. Heterocycl. Chem., 2018, 55(1), 258-264.

Reviewer 2 Report

Comments and Suggestions for Authors

Comments to Authors

In the current review, the authors have discussed an overview on the structure-activity relationship in novel antimicrobial thiazoles clubbed with various heterocycles (2017-2022). The authors have made decent efforts to come up with a review article. In the reviewer’s opinion, the manuscript has been organized appropriately. The reviewer feels that a potential reader can easily understand the details by consulting the main data outlined in the current version of the manuscript with few exceptions. To me, the write-up is clear and to the point which is crucial for an article with few exceptions which are acceptable. Although the same author has also published similar articles closely related to this piece of writing in general, the current version of the manuscript can bring another pile-up of existing literature pertaining to the Structure-Activity Relationship in Novel Antimicrobial Thiazoles Clubbed with Various Heterocycles, and the authors have also managed to organize a decent bulk of the relevant material. There seems to be a sequence rather than a random description of reports which may benefit the readership of this journal. Herein, I would like to recommend a publication of the current version. However, to this end, there is one suggestion related to the manuscript and authors should take into account this suggestion before resubmission.

Minor Comment

It would be great if the author could add a Figure of clinically approved drugs with IC50 values etc.,  bearing one thiazole ring well-known for antimicrobial drugs.

Reviewer 3 Report

Comments and Suggestions for Authors

The present review manuscript entitled “An Overview on the Structure-Activity Relationship in Novel Antimicrobial Thiazoles Clubbed with Various Heterocycles (2017-2022)" by Ungureanu et al involved the bibliographical analyses of current data (2017-2022) regarding the SAR In antimicrobial thiazole compounds.

The introduction is almost complete and according to the developed topic of the manuscript. Considering the submission date, it should be necessary to include the references of 2023 and amplify the years of study.

Taking into account this point, it should be important to clarify why the authors selected the specific range of years of study. Besides, it is important to know why the authors selected only the reported heterocycles (the title should be more specific about this point, which kind of heterocycles are included in this study?)

Also, the manuscript is interesting, organize, and focused on the topic that is of interest due to the current attention of antimicrobial potential agents.

Moreover, the information they described is supported with clear and logical figures and schemes that summarize all the required data for the discussion item. According to this point it should be appropriate to organize part of the information using a table to decrease the number of pages. By utilizing tables, the potential reader will understand the information in a simple and friendly way.

Furthermore, the following paragraphs should be improved “Based on the observations drawn in this review, it is easy to notice that each compound behaves differently based on its structure. However, they can exhibit the same pharmacological effect but with different potency” and “Thiazole clubbing with various heterocycles had different results depending on the 1346 other heterocycle(s)” because it is a common conclusion for each family or organic compound working on the medicinal chemistry field. Also, considering the amount of data the authors reported, the conclusions should be improved with the aim to select or decide about potential thiazole derivatives containing heterocycles moieties.

Finally, I would like to invite the authors to include the abbreviation list of words at the end of this manuscript.

Reviewer 4 Report

Comments and Suggestions for Authors The manuscript by Ungureanu et al. provides a solid review on the antimicrobial activity and the structure-activity relationship of series of hybrid compounds with the thiazole ring linked to different heterocycles as the scaffold of the hybrid molecules. In the manuscript, the authors pointed out the necessity of developing novel antimicrobials. The authors then introduced thiazole as an important scaffold in the development of novel antimicrobials. This is followed by a detailed review on the structure-activity relationship studies on different series of thiazole-based hybrid compounds.   Overall, this manuscript provides an informative review on the antimicrobial activity of thiazole-based hybrid compounds and could be useful for others in the field. I would recommend the publication of this manuscript after the following issues have been addressed:   1. The authors should be aware that disk diffusion method is not an accurate or reliable method to quantify the antimicrobial activity of compounds as different compounds have different diffusion rates on agar plates. A compound with a lower antimicrobial activity but a faster diffusion rate on agar plate could turn out showing a larger zone of inhibition than a compound with a higher antimicrobial activity but a slower diffusion rate. MIC values should be used to quantitatively compare the antimicrobial activity of compounds where possible.   2. For some series of compounds, the authors proposed a possible target or mechanism of action of compounds via docking studies. The authors should note that results from docking studies do not always correlate with the experimental results. If the proposed target or mechanism has been verified in biological assays, the results should be included in the manuscript. Otherwise, the authors should mention that the proposed target or mechanism has not been verified in biological assay.   3. SAR studies in this review mainly focus on comparing the MIC values and/or zone of inhibition of compounds. The authors should also include the SAR studies on the effects of compounds on different virulence factors (e.g. biofilm inhibition), in vivo experimental results and/or cytotoxicity of compounds where possible.   4. Minor comment: Numbers in R1, R2, R3 etc. should be in superscript, not subscript.

Round 2

Reviewer 3 Report

Comments and Suggestions for Authors

The authors performed all the suggested corrections, and they replied all the questions I made, thanks.